# QUERY EFFICIENT NONSMOOTH STOCHASTIC BLACK-BOX BILEVEL OPTIMIZATION WITH BREGMAN DISTANCE

## ABSTRACT

Bilevel optimization (BO) has recently gained significant attention in various machine learning applications due to its ability to model the hierarchical structures inherent in these problems. Several gradient-free methods have been proposed to address stochastic black-box bilevel optimization problems, where the gradients of both the upper and lower-level objective functions are unavailable. However, these methods suffer from high query complexity and do not accommodate more general bilevel problems involving nonsmooth regularization. In this paper, we present a query-efficient method that effectively leverages Bregman distance to solve nonsmooth stochastic black-box bilevel optimization problems. More importantly, we provide a non-asymptotic convergence analysis, showing that our method requires only $\mathcal{O}(d_1(d_1 + d_2)^2 \epsilon^{-2})$ queries to reach the $\epsilon$-stationary point. Additionally, we conduct experiments on data hyper-cleaning and hyper-representation learning tasks, demonstrating that our algorithms outperform existing bilevel optimization methods.

## 1 INTRODUCTION

Bilevel optimization (BO) (Bard, 2013; Colson et al., 2007) plays a central role in various significant machine learning applications, including hyper-parameter optimization (Pedregosa, 2016; Bergstra et al., 2011; Bertsekas, 1976; Shi & Gu, 2021; Shi et al., 2024), meta-learning (Feurer et al., 2015; Franceschi et al., 2018; Rajeswaran et al., 2019), reinforcement learning (Hong et al., 2023; Konda & Tsitsiklis, 2000). Generally speaking, the BO can be formulated as follows,

$$\min_{\mathbf{x} \in \mathcal{X}} \Phi(\mathbf{x}) = f(\mathbf{x}, \mathbf{y}^*(\mathbf{x})) + h(\mathbf{x}) \; s.t. \; \mathbf{y}^*(\mathbf{x}) = \arg\min_{\mathbf{y} \in \mathcal{Y}} g(\mathbf{x}, \mathbf{y}),$$

where $\mathcal{X}$ and $\mathcal{Y}$ are convex subsets in $\mathbb{R}^{d_1}$ and $\mathbb{R}^{d_2}$, respectively; $f$ is smooth and possibly nonconvex; $g$ is smooth and strongly convex; $h(\mathbf{x})$ is convex and possibly nonsmooth. This problem involves a competition between two parties or two objectives, and if one party makes its choice first, it will affect the optimal choice of the other party.

Recently, hypergradient methods have shown great effectiveness in solving various white-box bilevel optimization problems. Specifically, Franceschi et al. (2017); Pedregosa (2016); Ji et al. (2021) proposed the double-loop algorithms, which use the gradient methods to approximate the solution to the lower-level problem and then use the implicit differentiable methods (Pedregosa, 2016; Ji et al., 2021) or explicit differentiable methods (Franceschi et al., 2017) to approximate the gradient of the upper-level objective w.r.t $x$ to update $x$. However, in some real-world applications, such as in a sequential game, the problems must be updated simultaneously, which makes these methods unsuitable. To solve this problem, Hong et al. (2023); Khanduri et al. (2021); Chen et al. (2021b); Guo et al. (2021a); Shi et al. (2024) designed the single-loop methods, which use the Neumann series to approximate the hypergradient and then update $\mathbf{x}$ and $\mathbf{y}$ simultaneously. To further improve the performance, Dagréou et al. (2022); Yang et al. (2023); Chu et al. (2024); Huang (2024) proposed a fully single-loop method by introducing an auxiliary variable.

In many real-world applications, both the upper- and lower-level objectives may be black-box functions where gradient information is unavailable. Consequently, the abovementioned methods

cannot be directly applied to solve such black-box bilevel optimization problems. Researchers have increasingly focused on developing approaches to address this challenge in recent years, as shown in Table 1. Gu et al. (2021) use the Gaussian convolution approach for the upper-level function while assuming a good solution to the lower-level problem is available through an optimization algorithm such as the zeroth-order (proximal) gradient method. However, no convergence analysis is provided. Aghasi & Ghadimi (2024) use Gaussian smoothing on both upper and lower-level objectives and propose a zeroth-order hypergradient estimation of hypergradient by solving a linear system. In addition, the authors also show their method needs $\mathcal{O}(\frac{(d_1+d_2)^4}{\epsilon^3} \log \frac{d_1+d_2}{\epsilon})$ queries to obtain the stationary point of the original problem (i.e., $\|\nabla F(\mathbf{x})\|^2 \leq \epsilon$, where $F(\mathbf{x}) = f(\mathbf{x}, \mathbf{y}^*(\mathbf{x}))$). However, this method needs to solve the lower-level problem using the zeroth-order gradient method, which can result in high query complexity. Additionally, it is limited to addressing only the specific case of stochastic bilevel optimization where $h(\mathbf{x}) = 0$. Thus, developing methods to tackle the nonsmooth black-box bilevel optimization with reduced query requirements remains an open challenge.

Table 1: Summary of stochastic black-box bilevel optimization methods (The fourth and fifth columns show the property of the upper and lower-level problem e.g., S denotes smooth, NS denotes nonsmooth, NC denotes nonconvex, SC denotes strongly convex; *in the fifth column, we ignore the loop of approximating the zeroth-order gradient of all the methods*; the last column shows the query complexity to achieve the stationary point $\|\nabla F(\mathbf{x})\|^2 \leq \epsilon$ or $\frac{1}{\alpha}\|\mathbf{x}^{t+1} - \mathbf{x}^t\|^2 \leq \epsilon$).

| Methods | Ref. | UL | LL | Loops | Query Complexity |
|---------|------|-----|-----|-------|------------------|
| ZDSBA | Aghasi & Ghadimi (2024) | S,NC | SC | Double | $\mathcal{O}(\frac{(d_1+d_2)^4}{\epsilon^3} \log \frac{d_1+d_2}{\epsilon})$ |
| HOZOG | Gu et al. (2021) | S,NC | NC | Triple | $\times$ |
| BreSB$^3$O | Ours | NS,NC | SC | Single | $\mathcal{O}(\frac{d_1(d_1+d_2)^2}{\epsilon^2})$ |

To fill this gap, in the paper, we propose a query-efficient bilevel optimization method to solve the nonsmooth stochastic black-box bilevel optimization problem. Specifically, we first approximate the bilevel optimization by using the Gaussian smoothing and then propose a fully single-loop framework with the zeroth-order gradient method to solve the new problem. We incorporate the Bregman distance and apply the mirror descent on the upper-level variables to solve the nonsmooth term. Theoretically, we present our method can converge to the stationary point of the original problem with $\mathcal{O}(\frac{d_1(d_1+d_2)^2}{\epsilon^2})$ queries. We conduct experiments on data hyper-cleaning and hyper-representation learning tasks to demonstrate that our new algorithms outperform related bilevel optimization approaches.

## 2 RELATED WORKS

### 2.1 HYPERGRADIENT BASED BILEVEL OPTIMIZATION

Bilevel optimization Bracken & McGill (1973) has been extensively studied for decades. Various strategies have emerged in recent years for addressing stochastic bilevel optimization problems. The first class of methods relies on a two-loop structure: an inner loop solves the lower-level problem, and an outer loop updates the upper-level variable using an approximate stochastic hypergradient. Bracken & McGill (1973); Chen et al. (2021a); Ji et al. (2021) employed SGD updates for the lower-level problem, where each outer-loop iteration estimates the Hessian-inverse-vector product by solving a linear system or use the explicit gradient method. Another class of methods focuses on single-loop algorithms, which alternately/simultaneously update the upper-level and lower-level variables. Specifically, Hong et al. (2023) introduced a method that uses Neumann approximations for the inverse Hessian, coupled with a single SGD step for the lower-level problem. Shi et al. (2024) proposed a new definition of the stationary point and proposed a new method for the lower-level constrained bilevel optimization problem. Guo et al. (2021b); Yang et al. (2021); Khanduri et al. (2021) incorporated momentum acceleration to improve convergence rates. However, these methods are not truly single-loop, as estimating the Hessian-inverse-vector product via the Neumann series requires an additional iterative subroutine. More recently, Dagréou et al. (2022) proposed a novel framework for bilevel optimization that allows simultaneous updates of the lower-level variables, the

linear system loss variables, and the upper-level variables. However, its applicability to black-box bilevel optimization has not yet been explored.

## 2.2 BREGMAN DISTANCE-BASED METHODS

The Bregman distance-based method (Bregman, 1967; Censor & Lent, 1981; Censor & Zenios, 1992; Beck & Teboulle, 2003) is an approach commonly used in optimization and machine learning, particularly in settings where traditional Euclidean distances are not the most suitable measure of divergence or similarity between points. This method leverages Bregman divergences, a generalization of distance measures tailored for convex functions and widely applied in problems such as convex optimization, iterative algorithms, and regularization techniques. Recently, the mirror descent method has been extended to address minimax or bilevel optimization challenges. Specifically, Babanezhad & Lacoste-Julien (2020) proposed an algorithm rooted in mirror descent principles for tackling convex-concave minimax scenarios. Rafique et al. (2022) designed a series of mirror descent strategies suitable for minimax optimization in weakly convex contexts. Additionally, a novel approach based on mirror descent principles (Mertikopoulos et al., 2019) has emerged for addressing certain nonconvex-nonconcave minimax issues characterized by non-monotonic variational inequalities. More recently, Huang et al. (2022) proposed a class of enhanced bilevel optimization methods based on Bregman distance to solve the nonconvex-strongly-convex stochastic bilevel optimization problems. In addition to first-order algorithms, some existing works focus on zeroth-order algorithms for solving minimax/mini optimization problems. For example, Paul et al. (2023) proposed almost sure convergence of zeroth-order mirror descent algorithm and can find an $\epsilon$-stationary point with the total complexity of $\mathcal{O}(\epsilon^{-2})$. Maheshwari et al. (2022) proposed a gradient descent-ascent algorithm with random reshuffling under convex-concave setting and obtained an $\epsilon$-stationary point with the total complexity of $\mathcal{O}(\epsilon^{-4})$. Under the nonconvex-concave setting, Xu et al. (2024) proposed the ZO-AGP algorithm and its iteration complexity of obtaining an $\epsilon$-staitionary point is bounded by $\mathcal{O}(\epsilon^{-4})$. However, the Bregman distance-based method in the black-box bilevel optimization has not been fully discussed.

## 3 PRELIMINARIES

### 3.1 NOTATIONS

Here, we give several important notations used in this paper. $\|\cdot\|$ denotes the $\ell_2$ norm for vectors and spectral norm for matrices. $I_d$ denotes a $d$-dimensional identity matrix. $\|\cdot\|_1$ denotes the $\ell_1$ norm for vectors. $A^\top$ denotes transpose of matrix $A$. Given a convex set $\mathcal{X}$, we define a projection operation to $\mathcal{X}$ as $\mathcal{P}_\mathcal{X}(x') = \arg\min_{x\in\mathcal{X}} 1/2\|x - x'\|^2$.

### 3.2 PROBLEM SETTING

In this paper, we study the following stochastic bilevel optimization problem:

$$\min_{\mathbf{x}\in\mathcal{X}\subseteq\mathbb{R}^{d_1}} f(\mathbf{x}, \mathbf{y}^*(\mathbf{x})) + h(\mathbf{x}) = \mathbb{E}_\xi[f(\mathbf{x}, \mathbf{y}^*(\mathbf{x}); \xi)] + h(\mathbf{x}), \tag{1}$$

$$\text{s.t. } \mathbf{y}^*(\mathbf{x}) \in \arg\min_{\mathbf{y}\in\mathbb{R}^{d_2}} g(\mathbf{x}, \mathbf{y}) = \mathbb{E}_\zeta[g(\mathbf{x}, \mathbf{y}; \zeta)],$$

where function $F(\mathbf{x}) = f(\mathbf{x}, \mathbf{y}^*(\mathbf{x}))$ is smooth and possibly nonconvex, and function $h(\mathbf{x})$ is convex and possibly nonsmooth, and function $g(\mathbf{x}, \mathbf{y})$ is $\mu_g$-strongly convex on $\mathbf{y}$.

Then, we introduce several mild assumptions on the Problem 1.

**Assumption 1.** *$f$ and $\nabla f$ are Lipschitz continuous in $(\mathbf{x}, \mathbf{y}) \in \mathcal{X} \times \mathbb{R}^{d_2}$ with Lipschitz constant $L_0^f$ and $L_1^f$;*

**Assumption 2.** *(1) $g, \nabla g$ and $\nabla^2 g$ are $L_0^g$, $L_1^g$ and $L_2^g$ Lipschitz continuous in $(\mathbf{x}, \mathbf{y})$, respectively; (2) $g(\mathbf{x}, \cdot)$ is $\mu_g$-strongly convex on $\mathbf{y}$ for any given $\mathbf{x} \in \mathcal{X}$.*

**Assumption 3.** *The function $h(\mathbf{x})$ is convex but possibly nonsmooth for any $\mathbf{x} \in \mathcal{X}$.*

**Assumption 4.** *Function $\Phi(\mathbf{x}) = F(\mathbf{x}) + h(\mathbf{x})$ is bounded, i.e., $\Phi = \inf_{\mathbf{x}\in\mathbb{R}^{d_1}} \Phi(\mathbf{x}) > -\infty$.*

**Assumption 5.** *In the general expectation setting, there exist positive constant $\sigma_f$, $\sigma_{g1}$ and $\sigma_{g2}$ such that $\mathbb{E}\left[\|\nabla f(\mathbf{x}, \mathbf{y}; \xi) - \nabla f(\mathbf{x}, \mathbf{y})\|^2\right] \leq \sigma_f^2$, $\mathbb{E}\left[\|\nabla g(\mathbf{x}, \mathbf{y}; \zeta) - \nabla g(\mathbf{x}, \mathbf{y})\|^2\right] \leq \sigma_{g,1}^2$ and $\mathbb{E}\left[\|\nabla^2 g(\mathbf{x}, \mathbf{y}; \zeta) - \nabla^2 g(\mathbf{x}, \mathbf{y})\|^2\right] \leq \sigma_{g,2}^2$.*

All these assumptions are commonly used in bilevel optimization problems (Ghadimi & Wang, 2018; Hong et al., 2023; Ji et al., 2021; Khanduri et al., 2021; Shi & Gu, 2021; Shi et al., 2022; Huang et al., 2022; Dagréou et al., 2022; Yang et al., 2023; Chu et al., 2024; Huang, 2024; Shi et al., 2024).

### 3.3 Gaussian Smoothing for Smooth Stochastic Black-Box Bilevel optimization

The traditional gradient-based methods for bilevel optimization need to calculate first and second-order gradients. However, upper and lower-level objective functions are black-box problems in this problem, and their gradient information is unavailable. To solve this problem, Aghasi & Ghadimi (2024) proposed a method using the zeroth-order gradient to approximate the gradient of upper and lower-level objective functions. Specifically, let $\eta = \{\eta_1, \eta_2\}$ and $\mu = \{\mu_1, \mu_2\}$ be two vectors in $\mathbb{R}_+^2$ and introduce the standard normal vectors $\mathbf{u} \sim \mathcal{N}(\mathbf{0}, I_{d_1})$ and $\mathbf{v} \sim \mathcal{N}(\mathbf{0}, I_{d_2})$, the Gaussian smooth approximation to Problem 1 without $h(\mathbf{x})$ can be formulated as

$$\min_{\mathbf{x} \in \mathbb{R}^{d_1}} F_{\eta\mu}(\mathbf{x}) = f_{\eta_1\mu_1}(\mathbf{x}, \mathbf{y}_{\eta_2\mu_2}^*(\mathbf{x})) = \mathbb{E}_{\mathbf{u}\mathbf{v}\xi}[f(\mathbf{x} + \eta_1\mathbf{u}, \mathbf{y}_{\eta_2\mu_2}^*(\mathbf{x}) + \mu_1\mathbf{v}; \xi)] \quad (2)$$

$$s.t. \ \mathbf{y}_{\eta_2\mu_2}^*(\mathbf{x}) = \arg\min_{\mathbf{y} \in \mathbb{R}^{d_2}} g_{\eta_2\mu_2}(\mathbf{x}, \mathbf{y}) = \mathbb{E}_{\mathbf{u}\mathbf{v}\zeta}[g(\mathbf{x} + \eta_2\mathbf{u}, \mathbf{y} + \mu_2\mathbf{v}; \zeta)].$$

Then, Aghasi & Ghadimi (2024) uses the zeroth-order gradient method to solve the lower-level problem. However, it is hard to get the (stochastic) gradient $\nabla F_{\eta\mu}(x)$ or $\nabla F_{\eta\mu}(x; \xi)$ since there is no closed form solution $\mathbf{y}_{\eta_2\mu_2}^*(\mathbf{x})$. To solve this problem, Aghasi & Ghadimi (2024) provides the gradient estimator of $\nabla F_{\eta\mu}(x)$ in the following lemma,

**Proposition 1.** *(Aghasi & Ghadimi (2024)) Under Assumptions 1 and 2, for any $\mathbf{x} \in \mathbb{R}^{d_1}$, we have*

$$\nabla F_{\eta\mu}(\mathbf{x}) = \nabla_1 f_{\eta_1\mu_1}(\mathbf{x}, \mathbf{y}_{\eta_2\mu_2}^*(\mathbf{x}))$$
$$- \nabla_{12}^2 g_{\eta_2\mu_2}(\mathbf{x}, \mathbf{y}_{\eta_2\mu_2}^*(\mathbf{x})) \left[\nabla_{22}^2 g_{\eta_2\mu_2}(\mathbf{x}, \mathbf{y}_{\eta_2\mu_2}^*(\mathbf{x}))\right]^{-1} \nabla_2 f_{\eta_1\mu_1}(\mathbf{x}, \mathbf{y}_{\eta_2\mu_2}^*(\mathbf{x})). \quad (3)$$

Since $\mathbf{y}_{\eta_2\mu_2}^*(\mathbf{x})$ is not available, Aghasi & Ghadimi (2024) use the results $\mathbf{y}_{\eta_2\mu_2}^T(\mathbf{x})$ of lower-level problem after $T$ iterations zeroth-order gradient descent as an estimation of $\mathbf{y}_{\eta_2\mu_2}^*(\mathbf{x})$. To efficiently approximate the zeroth-order Hessian inverse, Aghasi & Ghadimi (2024) solve the following linear system:

$$\mathbf{z}^*(\mathbf{x}) = \arg\min_{\mathbf{z}} \frac{1}{2} \left\langle \nabla_{22}^2 g_{\eta_2\mu_2}(\mathbf{x}, \mathbf{y}_{\eta_2\mu_2}^*(\mathbf{x}))\mathbf{z}, \mathbf{z}\right\rangle - \left\langle \nabla_2 f_{\eta_1\mu_1}(\mathbf{x}, \mathbf{y}_{\eta_2\mu_2}^*(\mathbf{x})), \mathbf{z}\right\rangle. \quad (4)$$

The key challenge lies in solving the lower-level problem, as performing $T$ iterations of zeroth-order gradient descent typically result in a high query count. Moreover, it focuses on smooth optimization problems, and its theoretical results cannot be directly extended to bilevel problems where the upper-level objective is nonsmooth.

## 4 Zeroth-Order Stochastic Bilevel Algorithm via Bregman Distance

In this section, we introduce our query efficient Zeroth-Order Stochastic Bilevel Algorithm via the Bregman distance (BreZOSBA) to solve the black-box nonsmooth stochastic BO problems. The algorithm is present in Algorithm 1.

### 4.1 Fully Single-Loop Zeroth-Order Gradient Method for Black-Box Bilevel Optimization

Before presenting our proposed method, we highlight key properties of $\mathbf{y}_{\eta_2\mu_2}^*(\mathbf{x})$ and $\mathbf{z}^*(\mathbf{x})$ in the following lemmas (detailed proof is available in our appendix), which play a crucial role in the design of our algorithm.

---

**Algorithm 1** BreZOSBA: Zeroth-Order Stochastic Bilevel Algorithm via Bregman Distance

---

**Input:** $\mathbf{x}^1 \in \mathcal{X} \subseteq \mathbb{R}^{d_1}, \mathbf{y}^1 \in \mathbb{R}^{d_2}, \mathbf{z}^1 \in \mathbb{R}^{d_2}$, number of iterations $T$, $r_{\mathbf{z}}$, batch size $B$ and $B'$, $0 < \alpha \leq 1$,
$\quad 0 < \beta \leq 1, 0 < \tau \leq 1$ and $0 < \lambda \leq 1$
1: **for** $t = 1, \ldots, T$ **do**
2: $\quad$ Compute $D_{\mathbf{x}}^t, D_{\mathbf{y}}^t, D_{\mathbf{z}}^t$.
3: $\quad$ Update $\mathbf{x}$: $\mathbf{x}^{t+1} = \arg\min_{\mathbf{x} \in \mathcal{X} \subseteq \mathbb{R}^{d_1}} \left\{ \langle D_{\mathbf{x}}^t, \mathbf{x} \rangle + h(\mathbf{x}) + \frac{1}{\alpha} \mathcal{B}_{\psi_t}(\mathbf{x}, \mathbf{x}^t) \right\}$.
4: $\quad$ Update $\mathbf{y}$: $\tilde{\mathbf{y}}^{t+1} = \mathbf{y}^t - \beta D_{\mathbf{y}}^t$, $\mathbf{y}^{t+1} = \mathbf{y}^t + \tau(\tilde{\mathbf{y}}^{t+1} - \mathbf{y}^t)$.
5: $\quad$ Update $\mathbf{z}$: $\mathbf{z}^{t+1} = \mathcal{P}_{\mathbf{z} \in \mathcal{Z}}(\mathbf{z}^t - \lambda D_{\mathbf{z}}^t)$.
6: **end for**

---

**Lemma 1.** *(Lipschitz continuity of $\mathbf{y}_{\eta_2 \mu_2}^*(\mathbf{x})$ and $\mathbf{z}^*(\mathbf{x})$) Under the Assumptions 1 and 2,
$\mathbf{y}^*(\mathbf{x})$ and $\mathbf{z}^*(\mathbf{x})$ are $L_{\mathbf{y}_{\eta\mu}^*}$ and $L_{\mathbf{z}^*}$-Lipschitz continuous, where $L_{\mathbf{y}_{\eta\mu}^*} = \frac{L_1^g}{\mu_g}$ and $L_{\mathbf{z}^*} = \left( \frac{L_1^f}{\mu_g} + \frac{L_0^f L_2^g}{\mu_g^2} \right) \left( 1 + \frac{L_1^g}{\mu_g} \right).$*

**Lemma 2.** *(Boundness of $\mathbf{z}^*(\mathbf{x})$) Under the Assumptions 1 and 2, $\mathbf{z}^*(\mathbf{x})$ is bounded by $r_{\mathbf{z}} = \frac{L_0^f}{\mu_g}$, i.e.,
$\|\mathbf{z}^*(\mathbf{x})\| \leq \frac{L_0^f}{\mu_g}.$*

Then, based on the above properties, we first show our query-efficient method for solving the Problem (2) where the nonsmooth term is ignored. Specifically, instead of using the results after $T$ iterations of the lower-level variable updating, we use $\mathbf{y}$ to replace the $\mathbf{y}_{\eta_2 \mu_2}^*(\mathbf{x})$ and define a practical hyper gradient estimator:

$$\bar{\nabla} f_{\eta\mu}(\mathbf{x}, \mathbf{y}) = \nabla_1 f_{\eta_1 \mu_1}(\mathbf{x}, \mathbf{y}) - \nabla_{12}^2 g_{\eta_2 \mu_2}(\mathbf{x}, \mathbf{y}) \left[ \nabla_{22}^2 g_{\eta_2 \mu_2}(\mathbf{x}, \mathbf{y}) \right]^{-1} \nabla_2 f_{\eta_1 \mu_1}(\mathbf{x}, \mathbf{y}). \quad (5)$$

To avoid calculating the Hessian inverse, one can solve the following linear system:

$$\mathbf{z}^*(\mathbf{x}, \mathbf{y}) := \left[ \nabla_{22}^2 g_{\eta_2 \mu_2}(\mathbf{x}, \mathbf{y}) \right]^{-1} \nabla_{\mathbf{y}} f_{\eta_1 \mu_1}(\mathbf{x}, \mathbf{y})$$

$$= \arg\min_{\mathbf{z} \in \mathcal{Z}} R(\mathbf{x}, \mathbf{y}, \mathbf{z}) = \arg\min_{\mathbf{z} \in \mathcal{Z}} \frac{1}{2} \left\langle \nabla_{22}^2 g_{\eta_2 \mu_2}(\mathbf{x}, \mathbf{y}) \mathbf{z}, \mathbf{z} \right\rangle - \left\langle \nabla_2 f_{\eta_1 \mu_1}(\mathbf{x}, \mathbf{y}), \mathbf{z} \right\rangle, \quad (6)$$

where $\mathcal{Z} = \left\{ \mathbf{z} \in \mathbb{R}^{d_2} | \|\mathbf{z}\| \leq r_{\mathbf{z}} \right\}$. This constraint is obtained from Lemma 2. Although this problem is relatively simple, obtaining solution $\mathbf{z}^*(\mathbf{x}, \mathbf{y})$ for each given $\mathbf{x}$ and $\mathbf{y}$ still takes high time complexity. Therefore, as in (Dagréou et al., 2022; Yang et al., 2023; Chu et al., 2024; Huang, 2024), we define the following hypergradient surrogates using $\mathbf{z}$ as an estimation of $\mathbf{z}^*(\mathbf{x}, \mathbf{y})$:

$$\bar{\nabla} f_{\eta\mu}(\mathbf{x}, \mathbf{y}, \mathbf{z}) = \nabla_1 f_{\eta_1 \mu_1}(\mathbf{x}, \mathbf{y}) - \nabla_{12}^2 g_{\eta_2 \mu_2}(\mathbf{x}, \mathbf{y}) \mathbf{z}, \quad (7)$$

where $\mathbf{z} \in \mathcal{Z}$ is an auxiliary vector to approximate the solution of Problem (6). Therefore, we can update $\mathbf{x}$, $\mathbf{y}$ and $\mathbf{z}$ at the same time for solving Problem (2):

$$\mathbf{x}^{t+1} = \mathbf{x}^t - \alpha D_{\mathbf{x}}^t, \ \mathbf{y}^{t+1} = \mathbf{y}^t - \beta D_{\mathbf{y}}^t, \ \mathbf{z}^{t+1} = \mathcal{P}_{\mathbf{z} \in \mathcal{Z}}(\mathbf{z}^t - \lambda D_{\mathbf{z}}^t), \text{ for } t \geq 1, \quad (8)$$

where $0 < \alpha \leq 1, 0 < \beta \leq 1, 0 < \lambda \leq 1$, $\bar{\xi}_i^t = [\xi_i^t, \zeta_i^t]$, $\bar{\mathbf{u}}_j^t = [\mathbf{u}_{1,j}^t, \mathbf{u}_{2,j}^t]$, $\bar{\mathbf{v}}_j^t = [\mathbf{v}_{1,j}^t, \mathbf{v}_{2,j}^t]$ and

$$D_{\mathbf{x}}^t = \frac{1}{BB'} \sum_{i=1}^B \sum_{j=1}^{B'} \bar{\nabla} f_{\eta\mu}(\mathbf{x}^t, \mathbf{y}^t, \mathbf{z}^t; \bar{\xi}_i^t, \bar{\mathbf{u}}_j^t, \bar{\mathbf{v}}_j^t)$$

$$= \frac{1}{BB'} \sum_{i=1}^B \sum_{j=1}^{B'} \left[ \hat{\nabla}_1 f_{\eta_1 \mu_1}(\mathbf{x}^t, \mathbf{y}^t; \xi_i^t, \mathbf{u}_{1,j}^t, \mathbf{v}_{1,j}^t) - \hat{\nabla}_{12}^2 g_{\eta_2 \mu_2}(\mathbf{x}^t, \mathbf{y}^t; \zeta_i^t, \mathbf{u}_{2,j}^t, \mathbf{v}_{2,j}^t) \mathbf{z}^t \right], \quad (9)$$

$$D_{\mathbf{y}}^t = \frac{1}{BB'} \sum_{i=1}^B \sum_{j=1}^{B'} \hat{\nabla}_2 g_{\eta_2 \mu_2}(\mathbf{x}^t, \mathbf{y}^t; \zeta_i^t, \mathbf{u}_{2,j}^t, \mathbf{v}_{2,j}^t), \quad (10)$$

$$D_{\mathbf{z}}^t = \frac{1}{BB'} \sum_{i=1}^B \sum_{j=1}^{B'} \nabla_{\mathbf{z}} R(\mathbf{x}^t, \mathbf{y}^t, \mathbf{z}^t; \bar{\xi}_i^t, \bar{\mathbf{u}}_j^t, \bar{\mathbf{v}}_j^t)$$

$$=\frac{1}{BB'}\sum_{i=1}^{B}\sum_{j=1}^{B'}\left[\hat{\nabla}_{22}^2 g_{\eta_2\mu_2}(\mathbf{x}^t,\mathbf{y}^t;\zeta_i^t,\mathbf{u}_{2,j}^t,\mathbf{v}_{2,j}^t)\mathbf{z}^t - \hat{\nabla}_2 f_{\eta_1\mu_1}(\mathbf{x}^t,\mathbf{y}^t;\xi_i^t,\mathbf{u}_{1,j}^t,\mathbf{v}_{1,j}^t)\right]. \quad (11)$$

Here, $B > 0$ and $B' > 0$ denote the batch sizes; $\hat{\nabla}_1 f_{\eta_1\mu_1}$, $\hat{\nabla}_2 f_{\eta_1\mu_1}$, $\hat{\nabla}_2 g_{\eta_2\mu_2}$, $\hat{\nabla}_{12}^2 g_{\eta_2\mu_2}$ and $\hat{\nabla}_{22}^2 g_{\eta_2\mu_2}$ are the stochastic zeroth-order gradient estimations defined as follows,

$$\hat{\nabla}_1 f_{\eta_1\mu_1}(\mathbf{x},\mathbf{y};\xi,\mathbf{u},\mathbf{v}) = \frac{1}{2\eta_1}\left[f^+(\mathbf{x},\mathbf{y};\xi) - f^-(\mathbf{x},\mathbf{y};\xi)\right]\mathbf{u}, \quad (12)$$

$$\hat{\nabla}_2 f_{\eta_1\mu_1}(\mathbf{x},\mathbf{y};\xi,\mathbf{u},\mathbf{v}) = \frac{1}{2\mu_1}\left[f^+(\mathbf{x},\mathbf{y};\xi) - f^-(\mathbf{x},\mathbf{y};\xi)\right]\mathbf{v}, \quad (13)$$

$$\hat{\nabla}_2 g_{\eta_2\mu_2}(\mathbf{x},\mathbf{y};\zeta,\mathbf{u},\mathbf{v}) = \frac{1}{2\mu_2}\left[g^+(\mathbf{x},\mathbf{y};\zeta) - g^-(\mathbf{x},\mathbf{y};\zeta)\right]\mathbf{v}, \quad (14)$$

$$\hat{\nabla}_{12}^2 g_{\eta_2\mu_2}(\mathbf{x},\mathbf{y};\zeta,\mathbf{u},\mathbf{v}) = \frac{1}{\eta_2\mu_2}\left[g^+(\mathbf{x},\mathbf{y};\zeta) + g^-(\mathbf{x},\mathbf{y};\zeta) - 2g(\mathbf{x},\mathbf{y};\zeta)\right]\mathbf{u}\mathbf{v}^\top, \quad (15)$$

$$\hat{\nabla}_{22}^2 g_{\eta_2\mu_2}(\mathbf{x},\mathbf{y};\zeta,\mathbf{u},\mathbf{v}) = \frac{1}{2\mu_2^2}\left[g^+(\mathbf{x},\mathbf{y};\zeta) + g^-(\mathbf{x},\mathbf{y};\zeta) - 2g(\mathbf{x},\mathbf{y};\zeta)\right](\mathbf{v}\mathbf{v}^\top - I_{d_2}), \quad (16)$$

where $f^+(\mathbf{x},\mathbf{y};\xi) = f(\mathbf{x}+\eta_1\mathbf{u},\mathbf{y}+\mu_1\mathbf{v};\xi)$, $f^-(\mathbf{x},\mathbf{y};\xi) = f(\mathbf{x}-\eta_1\mathbf{u},\mathbf{y}-\mu_1\mathbf{v};\xi)$, $g^+(\mathbf{x},\mathbf{y};\xi) = g(\mathbf{x}+\eta_2\mathbf{u},\mathbf{y}+\mu_2\mathbf{v};\zeta)$ and $g^-(\mathbf{x},\mathbf{y};\xi) = g(\mathbf{x}-\eta_2\mathbf{u},\mathbf{y}-\mu_2\mathbf{v};\zeta)$. Obviously, $D_{\mathbf{x}}^t$, $D_{\mathbf{y}}^t$ and $D_{\mathbf{z}}^t$ are unbiased estimations of $\bar{\nabla}f_{\eta\mu}(\mathbf{x}^t,\mathbf{y}^t,\mathbf{z}^t)$, $\nabla_2 g(\mathbf{x}^t,\mathbf{y}^t)$ and $\nabla_{\mathbf{z}}R(\mathbf{x}^t,\mathbf{y}^t,\mathbf{z}^t)$, i.e., $\mathbb{E}[D_{\mathbf{x}}^t] = \bar{\nabla}f(\mathbf{x}^t,\mathbf{y}^t,\mathbf{z}^t) = \nabla_1 f(\mathbf{x}^t,\mathbf{y}^t) - \nabla_{12}^2 g(\mathbf{x}^t,\mathbf{y}^t)\mathbf{z}^t$, $\mathbb{E}[D_{\mathbf{y}}^t] = \nabla_2 g(\mathbf{x}^t,\mathbf{y}^t)$, and $\mathbb{E}[D_{\mathbf{z}}^t] = \nabla_{\mathbf{z}}R(\mathbf{x}^t,\mathbf{y}^t,\mathbf{z}^t) = \nabla_{22}^2 g(\mathbf{x}^t,\mathbf{y}^t)\mathbf{z}^t - \nabla_2 f(\mathbf{x}^t,\mathbf{y}^t)$. In addition, we have the bounded variance of our batch zeroth-order estimations in the following lemma,

**Lemma 3.** *Under Assumption 5, the batch zeroth-order gradient estimations have bounded variance such that* $\mathbb{E}\left[\left\|D_{\mathbf{x}}^t - \bar{\nabla}f_{\eta\mu}(\mathbf{x},\mathbf{y},\mathbf{z})\right\|^2\right] \leq \frac{\sigma_{\mathbf{x}}^2}{BB'}$, $\mathbb{E}\left[\left\|D_{\mathbf{y}}^t - \nabla_2 g_{\eta_2\mu_2}(\mathbf{x},\mathbf{y})\right\|^2\right] \leq \frac{\sigma_{\mathbf{y}}^2}{BB'}$ *and* $\mathbb{E}\left[\left\|D_{\mathbf{z}}^t - \nabla_{\mathbf{z}}R(\mathbf{x},\mathbf{y},\mathbf{z})\right\|^2\right] \leq \frac{\sigma_{\mathbf{z}}^2}{BB'}$, *where* $\sigma_{\mathbf{x}}^2 > 0$, $\sigma_{\mathbf{y}}^2 > 0$ *and* $\sigma_{\mathbf{z}}^2 > 0$ *are defined in our Appendix.*

**Remark 1.** *Lemma 3 indicates that the bound of variance is related to $d_1$ and $d_2$. Setting $\eta_1 = \mu_1 = \mathcal{O}(\frac{\epsilon}{\sqrt{(d_1+d_2)^3}})$ and $\eta_2 = \mu_2 = \mathcal{O}(\frac{\epsilon}{\sqrt{(d_1+d_2)^4}})$, we have $\sigma_{\mathbf{x}}^2 = \mathcal{O}(d_1(d_1+d_2)^2)$, $\sigma_{\mathbf{y}}^2 = \mathcal{O}(d_1(d_1+d_2)^2)$ and $\sigma_{\mathbf{z}}^2 = \mathcal{O}(d_1(d_1+d_2)^2)$.*

### 4.2 MIRROR DESCENT FOR UPPER-LEVEL VARIABLES

In this subsection, we present our method based on Bregman distance for solving the following Gaussian smoothing approximation of the nonsmooth bilevel optimization problem.

$$\min_{\mathbf{x}\in\mathbb{R}^{d_1}} \Phi_{\eta\mu}(\mathbf{x}) = \mathbb{E}_{\mathbf{uv}\xi}[f(\mathbf{x}+\eta_1\mathbf{u},\mathbf{y}_{\eta_2\mu_2}^*(\mathbf{x})+\mu_1\mathbf{v};\xi)] + h(\mathbf{x}) \quad (17)$$

$$s.t.\ \mathbf{y}_{\eta_2\mu_2}^*(\mathbf{x}) = \arg\min_{\mathbf{y}\in\mathbb{R}^{d_2}} g_{\eta_2\mu_2}(\mathbf{x},\mathbf{y}) = \mathbb{E}_{\mathbf{uv}\zeta}[g(\mathbf{x}+\eta_2\mathbf{u},\mathbf{y}+\mu_2\mathbf{v};\zeta)].$$

Given a $\rho$-strongly convex and continuously-differentiable function $\psi(\mathbf{x})$, i.e., $\langle\mathbf{x}_1 - \mathbf{x}_2, \nabla\psi(\mathbf{x}_1) - \nabla\psi(\mathbf{x}_2)\rangle \geq \rho\|\mathbf{x}_1 - \mathbf{x}_2\|^2$, we can define a Bregman distance (Censor & Lent, 1981; Censor & Zenios, 1992) for any $\mathbf{x}_1, \mathbf{x}_2 \in \mathcal{X}$:

$$\mathcal{B}_{\psi_t}(\mathbf{x}_1,\mathbf{x}_2) = \psi_t(\mathbf{x}_1) - \psi_t(\mathbf{x}_2) - \langle\nabla\psi_t(\mathbf{x}_2),\mathbf{x}_1 - \mathbf{x}_2\rangle. \quad (18)$$

Then, for the nonsmooth bilevel optimization problem, we can replace the update rule of $\mathbf{x}$ in Eqn.(8) with the following mirror descent rule:

$$\mathbf{x}^{t+1} = \arg\min_{\mathbf{x}\in\mathcal{X}\subseteq\mathbb{R}^{d_1}}\left\{\langle D_{\mathbf{x}}^t,\mathbf{x}\rangle + h(\mathbf{x}) + \frac{1}{\alpha}\mathcal{B}_{\psi_t}(\mathbf{x},\mathbf{x}^t)\right\}. \quad (19)$$

Let $\psi(\mathbf{x}) = \frac{1}{2}\|\mathbf{x}\|^2$, we have $\mathcal{B}_{\psi_t}(\mathbf{x}_1,\mathbf{x}_2) = \frac{1}{2}\|x_1 - x_2\|^2$. If $\mathcal{X} = \mathbb{R}^{d_1}$, the Problem (19) is equivalent to the proximal gradient descent. If $\mathcal{X} \subseteq \mathbb{R}^{d_1}$ and $h(\mathbf{x}) = 0$, Problem (19) is equivalent to the projection gradient descent. When Bregman function $\psi_t(\mathbf{x},\mathbf{x}^t) = \frac{1}{2}(\mathbf{x}^t)^\top H_t\mathbf{x}^t$, we have $\mathcal{B}_{\psi_t}(\mathbf{x},\mathbf{x}^t) = \frac{1}{2}(\mathbf{x} - \mathbf{x}^t)^\top H_t(\mathbf{x} - \mathbf{x}^t)$. If $H_t$ is an adaptive matrix, Problem (19) is equivalent to the

proximal adaptive gradient decent. For example, we can generate the matrix $H_t$ like in Adam-type algorithms (Kingma, 2014).

In addition, instead of directly using the update rule of $\mathbf{y}$ in Eqn.(8), we introduce an addition parameter $0 < \tau \le 1$ and modify the update rule as follows

$$\tilde{\mathbf{y}}^{t+1} = \mathbf{y}^t - \beta D_{\mathbf{y}}^t, \quad \mathbf{y}^{t+1} = \mathbf{y}^t + \tau(\tilde{\mathbf{y}}^{t+1} - \mathbf{y}^t). \tag{20}$$

Note that this rule is used to derive the convergence result in the next section.

## 5 CONVERGENCE ANALYSIS

In this section, we discuss the convergence performance of our proposed BreZOSBA. All proofs are provided in our Appendix. We begin with introducing the metric $\|\mathcal{G}^t\|^2 \le \epsilon$ or $\mathbb{E}\|\mathcal{G}^t\|^2 \le \epsilon$ to evaluate the convergence performance (Huang et al., 2022; Ghadimi et al., 2016), where $\mathcal{G}^t$ is defined as: $\mathcal{G}^t = \frac{1}{\alpha}(\mathbf{x}^{t+1} - \mathbf{x}^t)$.

Then, we present several useful lemmas to derive our final result. (Detailed proofs are presented in our appendix.)

**Lemma 4.** *Under Assumptions 1, 2, and 5, setting $0 < \lambda \le \frac{1}{6L_2^g}$ and $\mathbf{z}_t^* = \mathbf{z}^*(\mathbf{x}^t, \mathbf{y}^t)$, we have*

$$\mathbb{E}\|\mathbf{z}^{t+1} - \mathbf{z}_{t+1}^*\|^2$$

$$\le \left(1 - \frac{\mu_g \lambda}{4}\right)\mathbb{E}[\|\mathbf{z}^t - \mathbf{z}_t^*\|^2] - \frac{3}{4}\mathbb{E}[\|\mathbf{z}^{t+1} - \mathbf{z}^t\|^2]$$

$$+ \frac{25\lambda\sigma_{\mathbf{z}}^2}{6\mu_g BB'} + \frac{20}{3}\left(\frac{L_1^f}{\mu_g} + \frac{L_0^f L_2^g}{\mu_g^2}\right)(\mathbb{E}[\|\mathbf{x}^{t+1} - \mathbf{x}^t\|^2] + \mathbb{E}[\|\mathbf{y}^{t+1} - \mathbf{y}^t\|^2]). \tag{21}$$

**Lemma 5.** *Under Assumptions 1, 2, 3, 4, and 5, setting $0 < \alpha \le \frac{3\rho}{4L_F}$, we have*

$$\Phi(\mathbf{x}^{t+1})$$

$$\le \Phi(\mathbf{x}^t) - \frac{3\alpha\rho}{8}\|\mathcal{G}^t\|^2 + \frac{16\alpha(L_1^g)^2}{\rho}\|\mathbf{z}^t - \mathbf{z}_t^*\|^2 + \frac{2\alpha}{\rho}A + \frac{4\alpha\sigma_{\mathbf{x}}^2}{\rho BB'}$$

$$+ \frac{2\alpha}{\rho}\left(8(L_1^f)^2 + 8(L_2^g r_{\mathbf{z}})^2 + 8(L_1^g)^2\left(\frac{2(L_1^f)^2}{\mu_g^2} + \frac{2(L_0^f L_2^g)^2}{\mu_g^4}\right)\right)\|\mathbf{y}^t - \mathbf{y}_{\eta_2\mu_2}^*(\mathbf{x}^t)\|^2, \tag{22}$$

*where $L_F = L_1^f + \frac{2L_1^f L_2^g + (L_0^f)^2 L_2^g}{\mu_g} + \frac{L_1^f (L_1^g)^2 + 2L_0^f L_1^g L_2^g}{\mu_g} + \frac{L_0^f (L_1^g)^2 L_2^g}{\mu_g^3}$ and $\sqrt{A} = L_1^f\sqrt{\frac{2L_1^g}{\mu_g}(\eta_2^2 d_1 + \mu_2^2 d_2)} + \frac{L_1^f}{2}(\eta_1(d_1 + 3)^{\frac{3}{2}} + \frac{\mu_1^2}{\eta_1}d_2 d_1^{\frac{1}{2}} + \frac{\eta_1^2}{\mu_1}d_1 d_2^{\frac{1}{2}} + \mu_1(d_2 + 3)^{\frac{3}{2}})$.*

**Lemma 6.** *Under Assumptions 1, 2, 3, 4, and 5, setting $0 < \beta \le \frac{1}{6L_2^g}$ and $\tau \le 1 \le \frac{1}{6\mu_g\beta}$ we have*

$$\mathbb{E}[\|\mathbf{y}^{t+1} - \mathbf{y}_{\eta_2\mu_2}^*(\mathbf{x}^{t+1})\|^2]$$

$$\le \left(1 - \frac{\mu_g \tau\beta}{4}\right)\mathbb{E}[\|\mathbf{y}^t - \mathbf{y}_{\eta_2\mu_2}^*(\mathbf{x}^t)\|^2] - \frac{3\tau}{4}\mathbb{E}[\|\tilde{\mathbf{y}}^{t+1} - \mathbf{y}^t\|^2] + \frac{25\beta\tau\sigma_{\mathbf{y}}^2}{6\mu_g BB'} + \frac{5L_{\mathbf{y}_{\eta\mu}^*}^2}{3}\mathbb{E}[\|\mathbf{x}^t - \mathbf{x}^{t+1}\|^2]$$

Using the above lemmas and assumptions, we can provide the following convergence performance of our method.

**Theorem 1.** *Under Assumptions 1, 2, 3, 4, and 5, setting $0 < \alpha \le \frac{3\rho}{4L_F}$, $0 < \beta \le \frac{1}{6L_2^g}$, $\tau \le 1 \le \frac{1}{6\mu_g\beta}$, $0 < \lambda \le \frac{1}{6L_2^g}$, and defining the following Lyapunov function for any $t \ge 1$: $H^t = \mathbb{E}[\Phi(\mathbf{x}^t)] + \mathbb{E}[\|\mathbf{y}^t - \mathbf{y}_{\eta_2\mu_2}^*(\mathbf{x}^t)\|^2] + \mathbb{E}[\|\mathbf{z}^t - \mathbf{z}_t^*\|^2]$, we have*

$$\frac{1}{T}\sum_{t=1}^{T}\mathbb{E}[\|\mathcal{G}^t\|^2] \le \frac{8H^1}{\alpha\rho T} + \frac{16}{\rho^2}\left(A + \frac{2\sigma_{\mathbf{x}}^2}{BB'}\right) + \frac{200\beta\tau\sigma_{\mathbf{y}}^2}{6\alpha\rho\mu_g BB'} + \frac{200\lambda\sigma_{\mathbf{z}}^2}{6\alpha\rho\mu_g BB'}. \tag{23}$$

**Remark 2.** *According to our lemmas, setting $\eta_1 = \mu_1 = \mathcal{O}(\frac{\epsilon}{\sqrt{(d_1+d_2)^3}})$ and $\eta_2 = \mu_2 = \mathcal{O}(\frac{\epsilon}{\sqrt{(d_1+d_2)^4}})$, we have $\sigma_{\mathbf{x}}^2 + \sigma_{\mathbf{y}}^2 + \sigma_{\mathbf{z}}^2 = \mathcal{O}(d_1(d_1 + d_2)^2)$ and $A = \mathcal{O}(\epsilon^2)$. If we take $BB' = \mathcal{O}(d_1(d_1 + d_2)^2\epsilon^{-1})$ and $T = \mathcal{O}(\epsilon^{-1})$, we have $\frac{1}{T}\sum_{t=1}^{T}\mathbb{E}[\|\mathcal{G}^t\|^2] \le \epsilon$. Therefore, the total query complexity of our method is $O(d_1(d_1 + d_2)^2\epsilon^{-2})$.*

## 6 EXPERIMENT

In this section, we evaluate the performance of our method in two applications: data hyper-cleaning and hyper-representation learning.

### 6.1 BASELINES

We compare our method with the following black-box bilevel methods:

1. **ZDSBA**: The method proposed in (Aghasi & Ghadimi, 2024) uses the Gaussian smoothing method to approximate upper and lower-level objectives and then solve a linear system to approximate the hypergradient.

2. **HOZOG**: The method proposed in (Gu et al., 2021) uses the zeroth-order gradient to solve the lower-level with perturbed upper-level variables and then use all the solutions to approximate the hypergradient.

We implement all the methods by Pytorch (Paszke et al., 2019) for hyper-representation learning and implement all the methods by Jax (Bradbury et al., 2018) and BenchmarkBilevel (Dagréou et al., 2022)for data hyper-cleaning. We run all the methods 5 times on a PC with four RTX3090 GPUs.

### 6.2 APPLICATIONS

Table 2: Test accuracy of all the methods on data hyper-cleaning

|  | MNIST | FashionMNIST | Cifar10 | SVHN |
|---|---|---|---|---|
| BreZOSBA | **86.42** ± 0.25 | **76.92** ± 0.19 | **38.08** ± 0.25 | **19.49** ± 0.45 |
| ZDSBA | 86.34 ± 0.17 | 76.92 ± 0.37 | 35.87 ± 0.15 | 16.15 ± 0.39 |
| HOZOG | 85.44 ± 0.47 | 76.29 ± 0.24 | 36.11 ± 0.03 | 16.98 ± 0.79 |

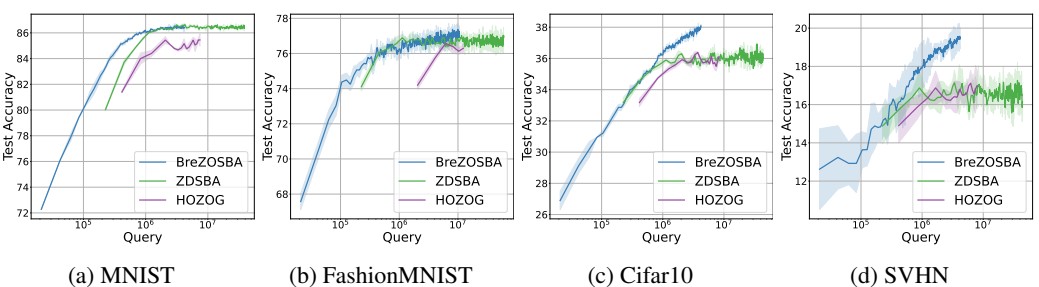

| (a) MNIST | (b) FashionMNIST | (c) Cifar10 | (d) SVHN |
|---|---|---|---|

Figure 1: Test accuracy against Queries of all the methods in hyper data-cleaning (We stop all the methods if the training time is more than 600 seconds).

**Data Hyper-cleaning.** In many real-world applications, the training and testing sets often follow different distributions, leading to performance discrepancies. To mitigate this issue, each data point is assigned an importance weight to bridge the gap between these distributions, which is referred to as data hyper-cleaning. This problem can be formulated as

$$\min_{\mathbf{x} \in \mathbb{R}^{d_1}} \sum_{\mathcal{D}_{val}} \ell\left(\mathbf{y}^*(\mathbf{x})^\top \mathbf{a}_i, \mathbf{b}_i\right) + c\|\mathbf{x}\|_1 \quad s.t. \quad \mathbf{y}^*(x) = \arg\min_{\mathbf{y} \in \mathbb{R}^{d_2}} \sum_{\mathcal{D}_{tr}} [\sigma(\mathbf{x})]_i \ell\left(\mathbf{y}^\top \mathbf{a}_i, \mathbf{b}_i\right),$$

where $c = 0.1$ is the regularization parameter, $\mathcal{D}_{tr}$ and $\mathcal{D}_{val}$ denote the training set and validation set respectively; $(\mathbf{a}_i, \mathbf{b}_i)$ denotes the data sample and label; $\sigma(\cdot) \coloneqq 1/(1 + exp(-\cdot))$ is the Sigmoid function; $\ell(\cdot, \cdot)$ is the loss function; $\ell(\cdot, \cdot)$ is the black-box loss function.

In this experiment, we compare our method with ZDSBA and HPZOG on datasets MNIST, FashionMNIST (Xiao et al., 2017), Cifar10, and SVHN. For ZDSBA and HOZOG, we set the inner iteration $T = 20$ and search the learning rate of upper and lower-level variables from the set $\{0.0001, 0.001, 0.01, 0.1, 1\}$. For our method, we set $r_\mathbf{z} = 10$, $\tau\beta = \lambda$ and choose $\lambda$ and $\alpha$ from the set $\{0.0001, 0.001, 0.01, 0.1, 1\}$. We set $\eta_1 = \eta_2 = \mu_1 = \mu_2 = 0.0001$ and $B = 64$ and $B' = 10$

Table 3: Test accuracy of all the methods on hyper-representation learning

|          | MNIST           | FashionMNIST    | Cifar10         | SVHN            |
| -------- | --------------- | --------------- | --------------- | --------------- |
| BreZOSBA | **89.31** ± 0.37 | **77.96** ± 0.31 | **22.81** ± 0.42 | 14.26 ± 0.86    |
| ZDSBA    | 85.76 ± 0.59    | 76.36 ± 0.10    | 20.90 ± 0.08    | **14.51** ± 1.62 |
| HOZOG    | 82.79 ± 1.08    | 75.11 ± 0.42    | 20.94 ± 0.44    | 14.48 ± 0.46    |

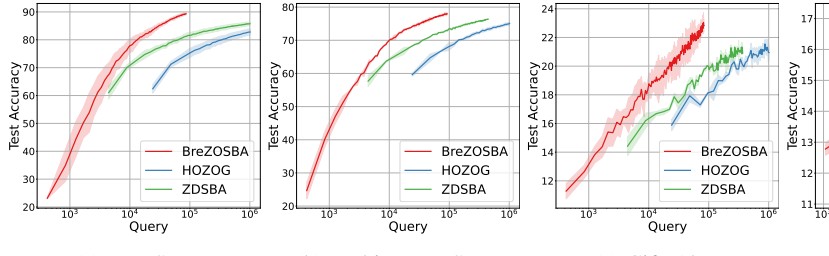

|     (a) MNIST     |  (b) FashionMNIST  |    (c) Cifar10    |     (d) SVHN     |

Figure 2: Test accuracy against Queries of all the methods in hyper-representation learning (We stop all the methods if the training time is more than 3600 seconds or the queries are more than $1e5$).

for all the methods. In this experiment, we run all the methods for 512 epochs. We stop all the methods if the running time exceeds 600 seconds.

**Hyper-representation Learning.** In this experiment, we perform the hyper-representation learning task over MNIST, FashionMNIST, Cifar10, and SVHN. The formulation of this problem is as follows:

$$\min_{\mathbf{x}\in\mathbb{R}^{d_1}} \sum_{\mathcal{D}_\mathcal{V}} \ell\left(\mathbf{y}^*(\mathbf{x})^\top \phi(\mathbf{x},\mathbf{a}_i), \mathbf{b}_i\right) + c_1\|\mathbf{x}\|_1 \ s.t. \ \mathbf{y}^*(\mathbf{x}) = \arg\min_{\mathbf{y}\in\mathbb{R}^{d_2}} \sum_{\mathcal{D}_\mathcal{T}} \ell\left(\mathbf{y}^\top \phi(\mathbf{x},\mathbf{a}_i), \mathbf{b}_i\right) + c_2\|\mathbf{y}\|^2,$$

where $\ell$ denotes the black-box loss function, $\mathcal{D}_\mathcal{V}$ and $\mathcal{D}_\mathcal{T}$ are training and validation datasets for randomly sampled meta task; $\phi(\mathbf{x},\cdot)$ is a neural network parameterized by $\mathbf{x}$, which denotes a representation mapping; $\mathbf{y}$ denotes the parameters of the classifier; $c_1 = 0.0001$ and $c_2 = 0.0001$ are the regularization parameters.

We use the LeNet-5 (LeCun et al., 1998) as $\phi$ to capture the features in this experiment. For ZDSBA and HOZOG, we set the inner iteration $T = 10$ and search the learning rate of upper and lower-level variables from the set $\{0.0001, 0.001, 0.01, 0.1, 1\}$. For our method, we set $r_\mathbf{z} = 10$, $\tau\beta = \lambda$ and choose $\lambda$ and $\alpha$ from the set $\{0.0001, 0.001, 0.01, 0.1, 1\}$. We set $\eta_1 = \eta_2 = \mu_1 = \mu_2 = 0.0001$, $B = 256$, and $B' = 5$ for all the methods. In this experiment, we run all the methods for 1000 epochs. We stop all the methods if the running time is larger than 3600 seconds or queries are larger than $100,000$.

## 6.3 RESULTS AND DISCUSSION

All the results are presented in Figure 1, 2 and Table 2, 3. Across all experiments, our method consistently demonstrates superior performance compared to ZDSBA and HOZOG, both in terms of test accuracy and query efficiency. This performance advantage is particularly evident when considering query costs, where our approach significantly reduces the number of queries required to achieve comparable or better accuracy. One key reason for this improvement is that HOZOG, while effective, depends on zeroth-order gradient approximations to compute the lower-level solution for each perturbed upper-level variable. This process inherently results in a much higher query demand, as zeroth-order methods are typically less query-efficient due to their reliance on sampling techniques. Similarly, ZDSBA employs a zeroth-order method for solving the lower-level problem, further exacerbating the query overhead and slowing convergence. In contrast, our method introduces a more query-efficient framework by reducing the reliance on expensive zeroth-order computations. The observed improvements in query efficiency and accuracy strongly support the theoretical convergence guarantees we established in Theorem 1. These findings not only validate the effectiveness of our approach but also highlight its potential for application in large-scale problems where query efficiency is critical.

Table 4: Test accuracy of our method on hyper data-cleaning with different $B'$

|  | MNIST | FashionMNIST | Cifar10 | SVHN |
|---|---|---|---|---|
| $B' = 1$ | $85.19 \pm 0.30$ | $75.04 \pm 0.63$ | $34.15 \pm 0.17$ | $15.64 \pm 0.54$ |
| $B' = 3$ | $86.34 \pm 0.16$ | $76.39 \pm 0.32$ | $36.64 \pm 0.42$ | $18.13 \pm 0.40$ |
| $B' = 5$ | $86.30 \pm 0.12$ | $76.73 \pm 0.41$ | $36.98 \pm 0.32$ | $18.85 \pm 0.52$ |
| $B' = 10$ | $\mathbf{86.42} \pm 0.25$ | $\mathbf{76.92} \pm 0.19$ | $\mathbf{38.08} \pm 0.25$ | $\mathbf{19.49} \pm 0.45$ |

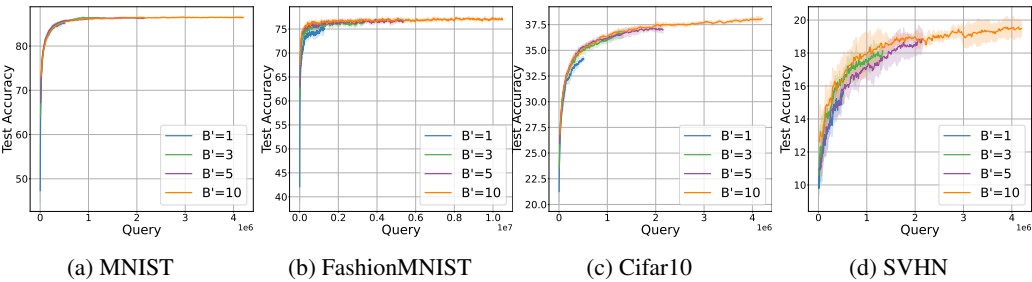

(a) MNIST      (b) FashionMNIST      (c) Cifar10      (d) SVHN

Figure 3: Test accuracy against Queries of our method in hyper data-cleaning with different $B'$.

## 6.4 Ablation Studies

We conduct ablation studies on the hyperparameters $B'$ and $r_{\mathbf{z}}$ to systematically evaluate their influence on performance. To maintain controlled conditions, we vary one hyperparameter while keeping all others fixed at their default values, ensuring a clear understanding of each parameter's effect. Additionally, we tune the step size for each configuration, as specified in the experimental setup, to optimize performance. The results of these experiments are presented in Tables 4, Figure 3, and Figure 4, with further supplementary results provided in the Appendix for additional insights.

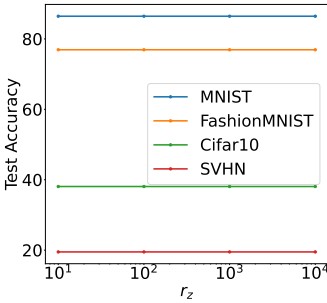

Figure 4: Test accuracy with different $r_{\mathbf{z}}$ of our method in hyper data-cleaning.

From the experiments, it is evident that increasing $B'$ consistently leads to both higher query requirements and improved test accuracy. This behaviour can be attributed to a larger $B'$ reducing the variance of gradient estimates, resulting in smoother updates and better overall convergence. Moreover, the improved convergence performance from a larger $B'$ suggests that the model can benefit from more stable gradients, facilitating more accurate learning, especially in scenarios with noisy or uncertain data. These findings highlight the importance of tuning $B'$ to balance queries and model accuracy. In addition, We can find that our method exhibits considerable stability to the parameter $r_{\mathbf{z}}$.

## 7 Conclusion

In this paper, we propose a query-efficient algorithm for stochastic black-box nonsmooth bilevel optimization, named BreZOSBA, and provide a comprehensive theoretical analysis. Our main result, Theorem 1, demonstrates that, under specific assumptions and hyperparameter settings, the output of BreZOSBA satisfies $\frac{1}{T} \sum_{t=1}^{T} \mathbb{E}[\|\mathcal{G}^t\|^2] \leq \epsilon$ with the query complexity of $\mathcal{O}\left(\frac{d_1(d_1+d_2)^2}{\epsilon^2}\right)$, which surpasses the performance of existing methods. We further validate the effectiveness of BreZOSBA through experiments on two applications, where our method consistently demonstrates superior performance.

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

# A  ADDITIONAL EXPERIMENTAL RESULTS

## A.1  IMPACT OF $B'$ AND $r_{\mathbf{z}}$

In this section, we show the results of our method using different $B'$ and $r_{\mathbf{z}}$ in hyper-representation learning in Figure 5 and Table 5 and Figure 6. We can find using a larger $B'$ can obtain a better performance and increase the query. We can find that our method exhibits considerable stability to the parameter $r_{\mathbf{z}}$.

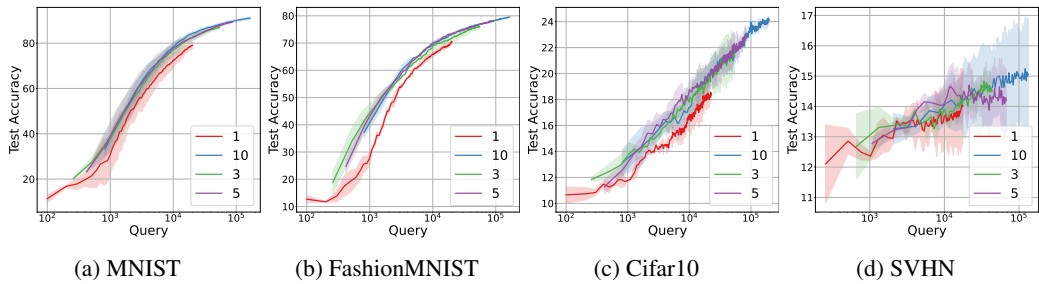

| (a) MNIST | (b) FashionMNIST | (c) Cifar10 | (d) SVHN |

Figure 5: Test accuracy against Queries of our method in hyper-representation learning using different $B'$.

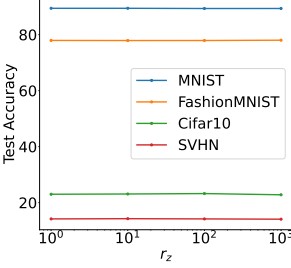

Figure 6: Test accuracy of our method in hyper-representation learning using different $\mu$.

Table 5: Test accuracy of our method on hyper-representation learning with different $B'$

|  | MNIST | FashionMNIST | Cifar10 | SVHN |
|---|---|---|---|---|
| $B' = 1$ | $78.96 \pm 1.45$ | $70.52 \pm 0.55$ | $18.37 \pm 0.23$ | $14.01 \pm 1.22$ |
| $B' = 3$ | $86.93 \pm 0.59$ | $76.05 \pm 0.93$ | $21.42 \pm 1.43$ | $14.51 \pm 0.36$ |
| $B' = 5$ | $89.31 \pm 0.37$ | $77.96 \pm 0.31$ | $22.81 \pm 0.42$ | $14.26 \pm 0.86$ |
| $B' = 10$ | $\mathbf{91.03} \pm 0.58$ | $\mathbf{79.57} \pm 0.08$ | $\mathbf{24.09} \pm 0.06$ | $\mathbf{15.06} \pm 1.81$ |

## A.2  IMPACT OF $\mu$

We conduct ablation studies on the hyperparameters $\eta_1 = \eta_2 = \mu_1 = \mu_2 = \mu$ to systematically evaluate their influence on performance. To maintain controlled conditions, we vary one hyperparameter while keeping all others fixed at their default values, ensuring a clear understanding of each parameter's effect. Additionally, we tune the step size for each configuration, as specified in the experimental setup, to optimize performance. We present the results in Table 6 and Table 7. We can find that our method exhibits considerable stability to the parameter $\mu$.

Table 6: Test accuracy of our method on data hyper-clean with different $\mu$

|  | MNIST | FashionMNIST | Cifar10 | SVHN |
|---|---|---|---|---|
| $\mu = 0.001$ | $85.32 \pm 0.21$ | $76.88 \pm 0.23$ | $38.11 \pm 0.34$ | $18.45 \pm 0.25$ |
| $\mu = 0.0001$ | $86.42 \pm 0.25$ | $76.92 \pm 0.19$ | $38.08 \pm 0.25$ | $19.49 \pm 0.45$ |
| $\mu = 0.00001$ | $86.45 \pm 0.18$ | $76.43 \pm 0.21$ | $38.07 \pm 0.21$ | $19.52 \pm 0.33$ |

Table 7: Test accuracy of our method on hyper-representation learning with different $\mu$

|  | MNIST | FashionMNIST | Cifar10 | SVHN |
|---|---|---|---|---|
| $\mu = 0.001$ | $88.32 \pm 0.15$ | $75.52 \pm 0.33$ | $21.11 \pm 1.34$ | $13.44 \pm 1.12$ |
| $\mu = 0.0001$ | $88.38 \pm 0.20$ | $75.74 \pm 0.26$ | $21.51 \pm 1.13$ | $14.26 \pm 0.86$ |
| $\mu = 0.00001$ | $88.43 \pm 0.16$ | $75.81 \pm 0.31$ | $21.46 \pm 0.78$ | $14.56 \pm 0.63$ |

## B  ADDITIONAL LEMMAS

**Lemma 7.** *Under Assumptions 1 and 2, we have (1) $f_{\eta\mu}$ and $\nabla f_{\eta\mu}$ are Lipschitz continuous in $(\mathbf{x}, \mathbf{y})$ with Lipschitz constant $L_0^f$ and $L_1^f$; (3) $\nabla g_{\eta_2\mu_2}$ and $\nabla^2 g_{\eta_2\mu_2}$ are $L_1^g$ and $L_2^g$ Lipschitz continuous in $(\mathbf{x}, \mathbf{y})$, respectively; (4) $g_{\eta_2\mu_2}$ is $\mu_g$-strongly convex on $\mathbf{y}$ for any given $\mathbf{x}$.*

*Proof.* The results can be easily obtained according to (Aghasi & Ghadimi, 2024). $\square$

**Lemma 8.** *(Smoothness of function $F$ (Chu et al., 2024)) Under Assumptions 1 and 2, the function $F$ is $L_F$-Lipschitz continuous, where*

$$L_F = L_1^f + \frac{2L_1^f L_2^g + (L_0^f)^2 L_2^g}{\mu_g} + \frac{L_1^f (L_1^g)^2 + 2L_0^f L_1^g L_2^g}{\mu_g} + \frac{L_0^f (L_1^g)^2 L_2^g}{\mu_g^3} \tag{24}$$

## C  DETAILED PROOF OF IMPORTANT LEMMAS

### C.1  PROOF OF LEMMA 1

*Proof.* Using definition $\partial \mathbf{y}_{\eta_2\mu_2}^*(\mathbf{x}) = [\nabla_{11}^2 g_{\eta_2\mu_2}(\mathbf{x}, \mathbf{y}_{\eta_2\mu_2}^*(\mathbf{x}))]^{-1} \nabla_{12}^2 g_{\eta_2\mu_2}(\mathbf{x}, \mathbf{y}_{\eta_2\mu_2}^*(\mathbf{x}))$ and Lemma 7, we have $L_{\mathbf{y}^*} = \frac{L_1^g}{\mu_g}$.

For $\mathbf{z}^*(\mathbf{x})$, we have

$$\|\mathbf{z}^*(\mathbf{x}_1) - \mathbf{z}^*(\mathbf{x}_2)\|$$
$$= \|[\nabla_{11}^2 g_{\eta_2\mu_2}(\mathbf{x}_1, \mathbf{y}_{\eta_2\mu_2}^*(\mathbf{x}_1))]^{-1} \nabla_2 f_{\eta_1\mu_1}(\mathbf{x}_1, \mathbf{y}_{\eta_2\mu_2}^*(\mathbf{x}_1)) - [\nabla_{11}^2 g_{\eta_2\mu_2}(\mathbf{x}_2, \mathbf{y}_{\eta_2\mu_2}^*(\mathbf{x}_2))]^{-1} \nabla_2 f_{\eta_1\mu_1}(\mathbf{x}_2, \mathbf{y}_{\eta_2\mu_2}^*(\mathbf{x}_2))\|$$
$$\leq \|[\nabla_{11}^2 g_{\eta_2\mu_2}(\mathbf{x}_1, \mathbf{y}_{\eta_2\mu_2}^*(\mathbf{x}_1))]^{-1} (\nabla_2 f_{\eta_1\mu_1}(\mathbf{x}_1, \mathbf{y}_{\eta_2\mu_2}^*(\mathbf{x}_1)) - \nabla_2 f_{\eta_1\mu_1}(\mathbf{x}_2, \mathbf{y}_{\eta_2\mu_2}^*(\mathbf{x}_2)))\|$$
$$\quad + \|([\nabla_{11}^2 g_{\eta_2\mu_2}(\mathbf{x}_1, \mathbf{y}_{\eta_2\mu_2}^*(\mathbf{x}_1))]^{-1} - [\nabla_{11}^2 g_{\eta_2\mu_2}(\mathbf{x}_2, \mathbf{y}_{\eta_2\mu_2}^*(\mathbf{x}_2))]^{-1}) \nabla_2 f_{\eta_1\mu_1}(\mathbf{x}_2, \mathbf{y}_{\eta_2\mu_2}^*(\mathbf{x}_2))\|$$
$$\leq \left(\frac{L_1^f}{\mu_g} + \frac{L_0^f L_2^g}{\mu_g^2}\right) \|(\mathbf{x}_1, \mathbf{y}_{\eta_2\mu_2}^*(\mathbf{x}_1))) - (\mathbf{x}_2, \mathbf{y}_{\eta_2\mu_2}^*(\mathbf{x}_2)))\|$$
$$\leq \left(\frac{L_1^f}{\mu_g} + \frac{L_0^f L_2^g}{\mu_g^2}\right) (\|\mathbf{x}_1 - \mathbf{x}_2\| + \|\mathbf{y}_{\eta_2\mu_2}^*(\mathbf{x}_1)) - \mathbf{y}_{\eta_2\mu_2}^*(\mathbf{x}_2)\|)$$
$$\leq \left(\frac{L_1^f}{\mu_g} + \frac{L_0^f L_2^g}{\mu_g^2}\right) \left(1 + \frac{L_1^g}{\mu_g}\right) \|\mathbf{x}_1 - \mathbf{x}_2\| \tag{25}$$

Then defining $L_{\mathbf{z}^*} = \left(\frac{L_1^f}{\mu_g} + \frac{L_0^f L_2^g}{\mu_g^2}\right) \left(1 + \frac{L_1^g}{\mu_g}\right)$ concludes the proof. $\square$

## C.2 Proof of Lemma 2

*Proof.* Using the definition of $\mathbf{z}^*(\mathbf{x})$, and Lemma 7, we can easily obtain

$$\|\mathbf{z}^*(\mathbf{x})\| \leq \frac{L_0^f}{\mu_g} = r_{\mathbf{z}} \tag{26}$$

□

# D Detailed Proof of Lemma 3

Here, we given the bound on the variance of the stochastic gradient estimation.

**Lemma 9.** *The stochastic zeroth-order gradients have bounded variance as follows*

$$\mathbb{E}\left[\left\|\bar{\nabla}f_{\eta\mu}(\mathbf{x},\mathbf{y},\mathbf{z};\bar{\xi}_i,\bar{\mathbf{u}}_j,\bar{\mathbf{v}}_j) - \bar{\nabla}f_{\eta\mu}(\mathbf{x},\mathbf{y},\mathbf{z})\right\|^2\right] \leq \sigma_{\mathbf{x}}^2 \tag{27}$$

$$\mathbb{E}\left[\left\|\hat{\nabla}_2 g_{\eta_2\mu_2}(\mathbf{x},\mathbf{y};\zeta_i,\mathbf{u}_{2,j},\mathbf{v}_{2,j}) - \nabla_2 g_{\eta_2\mu_2}(\mathbf{x},\mathbf{y})\right\|^2\right] \leq \sigma_{\mathbf{y}}^2 \tag{28}$$

$$\mathbb{E}\left[\left\|\nabla_{\mathbf{z}}R(\mathbf{x},\mathbf{y},\mathbf{z};\bar{\xi}_i,\bar{\mathbf{u}}_j,\bar{\mathbf{v}}_j) - \nabla_{\mathbf{z}}R(\mathbf{x},\mathbf{y},\mathbf{z})\right\|^2\right] \leq \sigma_{\mathbf{z}}^2 \tag{29}$$

*where* $\sigma_{\mathbf{x}}^2 = 2(L_1^f)^2(\eta_1^2(d_1+6)^3 + \frac{\mu_1^4}{\eta_1^2}d_1(d_2+4)^2) + 4(d_1+2)(\frac{\mu_1^2}{\eta_1^2}+1)(\sigma_f^2+(L_0^f)^2) + 16(L_2^g)^2[\frac{\eta_2^4}{\mu_2^2}(d_1+8)^4 + \frac{2\mu_2^4}{\eta_2^2}d_1(d_2+12)^3]r_{\mathbf{z}}^2 + 12(d_1+2)(\frac{\eta_2^2}{\mu_2^2}(d_1+4)+6+\frac{5\mu_2^2}{\eta_2^2}(d_2+2))(d_1+d_2)(\sigma_{g,2}^2+(L_1^g)^2)r_{\mathbf{z}}^2, \sigma_{\mathbf{y}}^2 = (L_1^g)^2(\mu_2^2(d_2+6)^3 + \frac{\eta_2^4}{\mu_2^2}d_2(d_1+4)^2) + 4(d_2+2)(\frac{\eta_2^2}{\mu_2^2}+1)(\sigma_{g,1}^2+(L_0^g)^2)$ *and* $\sigma_{\mathbf{z}}^2 = 2(L_1^f)^2(\mu_1^2(d_2+6)^3 + \frac{\eta_1^4}{\mu_1^2}d_2(d_1+4)^2) + 4(d_2+2)(\frac{\eta_1^2}{\mu_1^2}+1)(\sigma_f^2+(L_0^f)^2) + 4(L_2^g)^2(2\eta_1^2(d_1+16)^4 + \frac{\mu_1^6}{\eta_1^4}(d_2+6)^3(d_1+3))r_{\mathbf{z}}^2 + 6(d_1+6)(\frac{5}{2}(d_1+6) + \frac{3\mu_1^2}{\eta_1^2} + \frac{\mu_1^4}{2\eta_1^4}(d_2+2))(d_1+d_2)(\sigma_{g,2}^2+(L_1^g)^2)r_{\mathbf{z}}^2.$

*Proof.* We have

$$\mathbb{E}\left[\left\|\hat{\nabla}_2 g_{\eta_2\mu_2}(\mathbf{x},\mathbf{y};\zeta_i,\mathbf{u}_{2,j},\mathbf{v}_{2,j}) - \nabla_2 g_{\eta_2\mu_2}(\mathbf{x},\mathbf{y})\right\|^2\right]$$

$$=\mathbb{E}\left[\left\|\hat{\nabla}_2 g_{\eta_2\mu_2}(\mathbf{x},\mathbf{y};\zeta_i,\mathbf{u}_{2,j},\mathbf{v}_{2,j})\right\|^2\right] - \|\nabla_2 g_{\eta_2\mu_2}(\mathbf{x},\mathbf{y})\|^2$$

$$\leq\mathbb{E}\left[\left\|\hat{\nabla}_2 g_{\eta_2\mu_2}(\mathbf{x},\mathbf{y};\zeta_i,\mathbf{u}_{2,j},\mathbf{v}_{2,j})\right\|^2\right]$$

$$\leq(L_1^g)^2\left(\mu_2^2(d_2+6)^3 + \frac{\eta_2^4}{\mu_2^2}d_2(d_1+4)^2\right) + \frac{4\eta_2^2}{\mu_2^2}d_2\mathbb{E}\|\nabla_1 g(\mathbf{x},\mathbf{y};\zeta_i)\|^2 + 4(d_2+2)\mathbb{E}\|\nabla_2 g(\mathbf{x},\mathbf{y};\zeta_i)\|^2$$

$$\leq(L_1^g)^2\left(\mu_2^2(d_2+6)^3 + \frac{\eta_2^4}{\mu_2^2}d_2(d_1+4)^2\right) + 4(d_2+2)\left(\frac{\eta_2^2}{\mu_2^2}+1\right)\mathbb{E}\|\nabla g(\mathbf{x},\mathbf{y};\zeta_i)\|^2$$

$$\leq(L_1^g)^2\left(\mu_2^2(d_2+6)^3 + \frac{\eta_2^4}{\mu_2^2}d_2(d_1+4)^2\right) + 4(d_2+2)\left(\frac{\eta_2^2}{\mu_2^2}+1\right)(\sigma_{g,1}^2+(L_0^g)^2) \tag{30}$$

where the third inequality is due to Corollary 2.1 in (Aghasi & Ghadimi, 2024); the last inequality is due to Assumption 2 and 5 .

Then, according to the definition of $\bar{\nabla}f_{\eta\mu}(\mathbf{x},\mathbf{y},\mathbf{z};\bar{\xi}_i,\bar{\mathbf{u}}_j,\bar{\mathbf{v}}_j)$, we have

$$\mathbb{E}\left[\left\|\bar{\nabla}f_{\eta\mu}(\mathbf{x},\mathbf{y},\mathbf{z};\bar{\xi}_i,\bar{\mathbf{u}}_j,\bar{\mathbf{v}}_j) - \bar{\nabla}f_{\eta\mu}(\mathbf{x},\mathbf{y},\mathbf{z})\right\|^2\right]$$

$$=\mathbb{E}\left[\left\|\hat{\nabla}_1 f_{\eta_1\mu_1}(\mathbf{x},\mathbf{y};\xi_i,\mathbf{u}_{1,j},\mathbf{v}_{1,j}) - \hat{\nabla}_{12}^2 g_{\eta_2\mu_2}(\mathbf{x},\mathbf{y};\zeta_i,\mathbf{u}_{2,j},\mathbf{v}_{2,j})\mathbf{z} - \nabla_1 f_{\eta_1\mu_1}(\mathbf{x},\mathbf{y}) + \nabla_{12}^2 g_{\eta_2\mu_2}(\mathbf{x},\mathbf{y})\mathbf{z}\right\|^2\right]$$

$$\leq 2\mathbb{E}\left[\left\|\hat{\nabla}_1 f_{\eta_1\mu_1}(\mathbf{x},\mathbf{y};\xi_i,\mathbf{u}_{1,j},\mathbf{v}_{1,j}) - \nabla_1 f_{\eta_1\mu_1}(\mathbf{x},\mathbf{y})\right\|^2\right]$$

$$+ 2\mathbb{E}\left[\left\|\nabla_{12}^2 g_{\eta_2\mu_2}(\mathbf{x}, \mathbf{y})\mathbf{z} - \hat{\nabla}_{12}^2 g_{\eta_2\mu_2}(\mathbf{x}, \mathbf{y}; \zeta_i, \mathbf{u}_{2,j}, \mathbf{v}_{2,j})\mathbf{z}\right\|^2\right]$$

$$\tag{31}$$

For the first term, we have

$$\mathbb{E}\left[\left\|\hat{\nabla}_1 f_{\eta_1\mu_1}(\mathbf{x}, \mathbf{y}; \xi_i, \mathbf{u}_{1,j}, \mathbf{v}_{1,j}) - \nabla_1 f_{\eta_1\mu_1}(\mathbf{x}, \mathbf{y})\right\|^2\right]$$

$$\leq (L_1^f)^2\left(\eta_1^2(d_1 + 6)^3 + \frac{\mu_1^4}{\eta_1^2}d_1(d_2 + 4)^2\right) + 4(d_1 + 2)\left(\frac{\mu_1^2}{\eta_1^2} + 1\right)(\sigma_f^2 + (L_0^f)^2) \tag{32}$$

where the inequality is due to Assumption 1 and 5. For the second term, we have

$$\mathbb{E}\left[\left\|\nabla_{12}^2 g_{\eta_2\mu_2}(\mathbf{x}, \mathbf{y})\mathbf{z} - \hat{\nabla}_{12}^2 g_{\eta_2\mu_2}(\mathbf{x}, \mathbf{y}; \zeta_i, \mathbf{u}_{2,j}, \mathbf{v}_{2,j})\mathbf{z}\right\|^2\right]$$

$$\leq \mathbb{E}\left[\left\|\hat{\nabla}_{12}^2 g_{\eta_2\mu_2}(\mathbf{x}, \mathbf{y}; \zeta_i, \mathbf{u}_{2,j}, \mathbf{v}_{2,j})\mathbf{z}\right\|^2\right]$$

$$\leq 8(L_2^g)^2\left[\frac{\eta_2^4}{\mu_2^2}(d_1 + 8)^4 + \frac{2\mu_2^4}{\eta_2^2}d_1(d_2 + 12)^3\right]r_{\mathbf{z}}^2 + \left(\frac{6\eta_2^2}{\mu_2^2}(d_1 + 4)(d_1 + 2)\mathbb{E}\|\nabla_{11}^2 g(\mathbf{x}, \mathbf{y}; \zeta_i)\|_F^2\right.$$

$$\left. + 36(d_1 + 2)\mathbb{E}\|\nabla_{12}^2 g(\mathbf{x}, \mathbf{y}; \zeta_i)\|_F^2 + \frac{30\mu_2^2}{\eta_2^2}d_1(d_2 + 2)\mathbb{E}\|\nabla_{22}^2 g(\mathbf{x}, \mathbf{y}; \zeta_i)\|_F^2\right)r_{\mathbf{z}}^2$$

$$\leq 8(L_2^g)^2\left[\frac{\eta_2^4}{\mu_2^2}(d_1 + 8)^4 + \frac{2\mu_2^4}{\eta_2^2}d_1(d_2 + 12)^3\right]r_{\mathbf{z}}^2$$

$$+ 6(d_1 + 2)\left(\frac{\eta_2^2}{\mu_2^2}(d_1 + 4) + 6 + \frac{5\mu_2^2}{\eta_2^2}(d_2 + 2)\right)\mathbb{E}\|\nabla^2 g(\mathbf{x}, \mathbf{y}; \zeta_i)\|_F^2 r_{\mathbf{z}}^2$$

$$\leq 8(L_2^g)^2\left[\frac{\eta_2^4}{\mu_2^2}(d_1 + 8)^4 + \frac{2\mu_2^4}{\eta_2^2}d_1(d_2 + 12)^3\right]r_{\mathbf{z}}^2$$

$$+ 6(d_1 + 2)\left(\frac{\eta_2^2}{\mu_2^2}(d_1 + 4) + 6 + \frac{5\mu_2^2}{\eta_2^2}(d_2 + 2)\right)(d_1 + d_2)(\sigma_{g,2}^2 + (L_1^g)^2)r_{\mathbf{z}}^2 \tag{33}$$

where the third inequality is due to Proposition 2.5 b) in (Aghasi & Ghadimi, 2024); the last inequality is due to Assumption 2 and 5. Therefore, we have

$$\mathbb{E}\left[\left\|\bar{\nabla}f_{\eta\mu}(\mathbf{x}, \mathbf{y}, \mathbf{z}; \bar{\xi}_i, \bar{\mathbf{u}}_j, \bar{\mathbf{v}}_j) - \bar{\nabla}f_{\eta\mu}(\mathbf{x}, \mathbf{y}, \mathbf{z})\right\|^2\right] \leq \sigma_{\mathbf{x}}^2 \tag{34}$$

According to the definition $\nabla_{\mathbf{z}} R(\mathbf{x}, \mathbf{y}, \mathbf{z}; \bar{\xi}_i, \bar{\mathbf{u}}_j, \bar{\mathbf{v}}_j) = \hat{\nabla}_{22}^2 g_{\eta_2\mu_2}(\mathbf{x}, \mathbf{y}; \zeta_i, \mathbf{u}_{2,j}, \mathbf{v}_{2,j})\mathbf{z} - \hat{\nabla}_2 f_{\eta_1\mu_1}(\mathbf{x}, \mathbf{y}; \xi_i, \mathbf{u}_{1,j}, \mathbf{v}_{1,j})$, we have

$$\mathbb{E}\left[\left\|\nabla_{\mathbf{z}} R(\mathbf{x}, \mathbf{y}, \mathbf{z}; \bar{\xi}_i, \bar{\mathbf{u}}_j, \bar{\mathbf{v}}_j) - \nabla_{\mathbf{z}} R(\mathbf{x}, \mathbf{y}, \mathbf{z})\right\|^2\right]$$

$$= \mathbb{E}\left[\left\|\hat{\nabla}_{22}^2 g_{\eta_2\mu_2}(\mathbf{x}, \mathbf{y}; \zeta_i, \mathbf{u}_{2,j}, \mathbf{v}_{2,j})\mathbf{z} - \hat{\nabla}_2 f_{\eta_1\mu_1}(\mathbf{x}, \mathbf{y}; \xi_i, \mathbf{u}_{1,j}, \mathbf{v}_{1,j}) - \nabla_{22}^2 g_{\eta_2\mu_2}(\mathbf{x}, \mathbf{y})\mathbf{z} + \nabla_2 f_{\eta_1\mu_1}(\mathbf{x}, \mathbf{y})\right\|^2\right]$$

$$\leq 2\mathbb{E}\left[\left\|\hat{\nabla}_2 f_{\eta_1\mu_1}(\mathbf{x}, \mathbf{y}; \xi_i, \mathbf{u}_{1,j}, \mathbf{v}_{1,j}) - \nabla_2 f_{\eta_1\mu_1}(\mathbf{x}, \mathbf{y})\right\|^2\right]$$

$$+ 2\mathbb{E}\left[\left\|\nabla_{22}^2 g_{\eta_2\mu_2}(\mathbf{x}, \mathbf{y})\mathbf{z} - \hat{\nabla}_{22}^2 g_{\eta_2\mu_2}(\mathbf{x}, \mathbf{y}; \zeta_i, \mathbf{u}_{2,j}, \mathbf{v}_{2,j})\mathbf{z}\right\|^2\right] \tag{35}$$

For the first term, we have

$$\mathbb{E}\left[\left\|\nabla_2 f_{\eta_1\mu_1}(\mathbf{x}, \mathbf{y}) - \hat{\nabla}_2 f_{\eta_1\mu_1}(\mathbf{x}, \mathbf{y}; \xi_i, \mathbf{u}_{1,j}, \mathbf{v}_{1,j})\right\|^2\right]$$

$$\leq (L_1^f)^2\left(\mu_1^2(d_2 + 6)^3 + \frac{\eta_1^4}{\mu_1^2}d_2(d_1 + 4)^2\right) + 4(d_2 + 2)\left(\frac{\eta_1^2}{\mu_1^2} + 1\right)(\sigma_f^2 + (L_0^f)^2) \tag{36}$$

For the second term, we have

$$\mathbb{E}\left[\left\|\nabla_{22}^2 g_{\eta_2\mu_2}(\mathbf{x},\mathbf{y})\mathbf{z} - \hat{\nabla}_{22}^2 g_{\eta_2\mu_2}(\mathbf{x},\mathbf{y};\zeta_i,\mathbf{u}_{2,j},\mathbf{v}_{2,j})\mathbf{z}\right\|^2\right]$$

$$\leq \mathbb{E}\left[\left\|\hat{\nabla}_{22}^2 g_{\eta_2\mu_2}(\mathbf{x},\mathbf{y};\zeta_i,\mathbf{u}_{2,j},\mathbf{v}_{2,j})\mathbf{z}\right\|^2\right]$$

$$\leq 2(L_2^g)^2\left(2\eta_1^2(d_1+16)^4 + \frac{\mu_1^6}{\eta_1^4}(d_2+6)^3(d_1+3)\right)r_{\mathbf{z}}^2 + \left(\frac{15}{2}(d_1+6)^2\mathbb{E}\|\nabla_{11}^2 g(\mathbf{x},\mathbf{y};\zeta_i)\|_F^2\right.$$

$$\left.+ \frac{3\mu_1^2}{\eta_1^2}(3d_1+13)\mathbb{E}\|\nabla_{12}^2 g(\mathbf{x},\mathbf{y};\zeta_i)\|_F^2 + \frac{3\mu_1^4}{2\eta_1^4}(d_2+2)(d_1+3)\mathbb{E}\|\nabla_{22}^2 g(\mathbf{x},\mathbf{y};\zeta_i)\|_F^2\right)r_{\mathbf{z}}^2$$

$$\leq 2(L_2^g)^2\left(2\eta_1^2(d_1+16)^4 + \frac{\mu_1^6}{\eta_1^4}(d_2+6)^3(d_1+3)\right)r_{\mathbf{z}}^2$$

$$+ 3(d_1+6)\left(\frac{5}{2}(d_1+6) + \frac{3\mu_1^2}{\eta_1^2} + \frac{\mu_1^4}{2\eta_1^4}(d_2+2)\right)r_{\mathbf{z}}^2\mathbb{E}\|\nabla^2 g(\mathbf{x},\mathbf{y};\zeta_i)\|_F^2$$

$$\leq 2(L_2^g)^2\left(2\eta_1^2(d_1+16)^4 + \frac{\mu_1^6}{\eta_1^4}(d_2+6)^3(d_1+3)\right)r_{\mathbf{z}}^2$$

$$+ 3(d_1+6)\left(\frac{5}{2}(d_1+6) + \frac{3\mu_1^2}{\eta_1^2} + \frac{\mu_1^4}{2\eta_1^4}(d_2+2)\right)(d_1+d_2)(\sigma_{g,2}^2 + (L_1^g)^2)r_{\mathbf{z}}^2 \tag{37}$$

Therefore, we have

$$\mathbb{E}\left[\left\|\nabla_{\mathbf{z}} R(\mathbf{x},\mathbf{y},\mathbf{z};\bar{\xi}_i,\bar{\mathbf{u}}_j,\bar{\mathbf{v}}_j) - \nabla_{\mathbf{z}} R(\mathbf{x},\mathbf{y},\mathbf{z})\right\|^2\right] \leq \sigma_{\mathbf{z}}^2 \tag{38}$$

$\square$

Using the above lemma, we can easily obtain the results in Lemma 3.

*Proof.* Using the results in Lemma 9, we have bounded variance as follows

$$\mathbb{E}\left[\left\|D_{\mathbf{x}}^t - \bar{\nabla} f_{\eta\mu}(\mathbf{x},\mathbf{y},\mathbf{z})\right\|^2\right] \leq \frac{\sigma_{\mathbf{x}}^2}{BB'} \tag{39}$$

$$\mathbb{E}\left[\left\|D_{\mathbf{y}}^t - \nabla_2 g_{\eta_2\mu_2}(\mathbf{x},\mathbf{y})\right\|^2\right] \leq \frac{\sigma_{\mathbf{y}}^2}{BB'} \tag{40}$$

$$\mathbb{E}\left[\left\|D_{\mathbf{z}}^t - \nabla_{\mathbf{z}} R(\mathbf{x},\mathbf{y},\mathbf{z})\right\|^2\right] \leq \frac{\sigma_{\mathbf{z}}^2}{BB'} \tag{41}$$

$\square$

### D.1 Detailed Proof of Lemma 4

*Proof.* Since $\mathbf{z}_t^* = \mathbf{z}^*(\mathbf{x}^t,\mathbf{y}^t) = \left[\nabla_{22}^2 g_{\eta_2\mu_2}(\mathbf{x}^t,\mathbf{y}^t)\right]^{-1}\nabla_2 f_{\eta_1\mu_1}(\mathbf{x}^t,\mathbf{y}^t)$, we have

$$\|\mathbf{z}^{t+1} - \mathbf{z}_{t+1}^*\|^2$$

$$= \|\mathbf{z}^{t+1} - \mathbf{z}_t^* + \mathbf{z}_t^* - \mathbf{z}_{t+1}^*\|^2$$

$$= \|\mathbf{z}^{t+1} - \mathbf{z}_t^*\|^2 + 2\langle\mathbf{z}^{t+1} - \mathbf{z}_t^*, \mathbf{z}_t^* - \mathbf{z}_{t+1}^*\rangle + \|\mathbf{z}_t^* - \mathbf{z}_{t+1}^*\|^2$$

$$\leq (1 + \frac{\mu_g\lambda}{4})\mathbb{E}\|\mathbf{z}^{t+1} - \mathbf{z}_t^*\|^2 + (1 + \frac{4}{\mu\lambda})\|\mathbf{z}_t^* - \mathbf{z}_{t+1}^*\|^2, \tag{42}$$

where the first inequality follows from Young's inequality and Cauchy-Schwarz inequality; the second follows from our Algorithm 1.

First, we discuss the bounds of the first term. Since $R(\mathbf{x},\mathbf{y},\mathbf{z}) = \frac{1}{2}\langle\nabla_{22}^2 g_{\eta_2\mu_2}(\mathbf{x},\mathbf{y})\mathbf{z},\mathbf{z}\rangle - \langle\nabla_2 f_{\eta_1\mu_1}(\mathbf{x},\mathbf{y}),\mathbf{z}\rangle$, we can easily obtain $R(\mathbf{x},\mathbf{y},\mathbf{z})$ is $\mu_g$-strongly convex and $L_2^g$-smooth on $z$. Then, we have

$$R(\mathbf{x}^t,\mathbf{y}^t,\mathbf{z})$$

$$\geq R(\mathbf{x}^t, \mathbf{y}^t, \mathbf{z}^t) + \langle \nabla_{\mathbf{z}} R(\mathbf{x}^t, \mathbf{y}^t, \mathbf{z}^t), \mathbf{z} - \mathbf{z}^t \rangle + \frac{\mu_g}{2} \|\mathbf{z} - \mathbf{z}^t\|^2$$

$$= R(\mathbf{x}^t, \mathbf{y}^t, \mathbf{z}^t) + \langle D_{\mathbf{z}}^t, \mathbf{z} - \mathbf{z}^{t+1} \rangle + \langle \nabla_{\mathbf{z}} R(\mathbf{x}^t, \mathbf{y}^t, \mathbf{z}^t) - D_{\mathbf{z}}^t, \mathbf{z} - \mathbf{z}^{t+1} \rangle$$

$$+ \langle \nabla_{\mathbf{z}} R(\mathbf{x}^t, \mathbf{y}^t, \mathbf{z}^t), \mathbf{z}^{t+1} - \mathbf{z}^t \rangle + \frac{\mu_g}{2} \|\mathbf{z} - \mathbf{z}^t\|^2. \tag{43}$$

Using $R(\mathbf{x}, \mathbf{y}, \mathbf{z})$ is $L_2^g$-smooth on $z$, we have

$$R(\mathbf{x}^t, \mathbf{y}^t, \mathbf{z}^{t+1})$$

$$\leq R(\mathbf{x}^t, \mathbf{y}^t, \mathbf{z}_t) + \langle \nabla_{\mathbf{z}} R(\mathbf{x}^t, \mathbf{y}^t, \mathbf{z}^t), \mathbf{z}^{t+1} - \mathbf{z}^t \rangle + \frac{L_2^g}{2} \|\mathbf{z}^{t+1} - \mathbf{z}^t\|^2$$

$$\tag{44}$$

Combining the above inequalities, we can obtain

$$R(\mathbf{x}^t, \mathbf{y}^t, \mathbf{z})$$

$$\geq R(\mathbf{x}^t, \mathbf{y}^t, \mathbf{z}^{t+1}) + \langle D_{\mathbf{z}}^t, \mathbf{z} - \mathbf{z}^{t+1} \rangle + \langle \nabla_{\mathbf{z}} R(\mathbf{x}^t, \mathbf{y}^t, \mathbf{z}^t) - D_{\mathbf{z}}^t, \mathbf{z} - \mathbf{z}^{t+1} \rangle$$

$$+ \frac{\mu_g}{2} \|\mathbf{z} - \mathbf{z}^t\|^2 - \frac{L_2^g}{2} \|\mathbf{z}^{t+1} - \mathbf{z}^t\|^2 \tag{45}$$

According to the update rule $z^{t+1} = \mathcal{P}_{\mathcal{Z}}(z^t - \lambda D_{\mathbf{z}}^t) = \arg\min_{z \in \mathcal{Z}} \langle D_{\mathbf{z}}^t, \mathbf{z} - \mathbf{z}^t \rangle + \frac{1}{2\lambda} \|\mathbf{z} - \mathbf{z}^t\|^2$, where $\mathcal{Z} = \{\mathbf{z} \in \mathbb{R}^{d_2} | \|\mathbf{z}\| \leq r_{\mathbf{z}}\}$ we have

$$\langle D_{\mathbf{z}}^t + \frac{1}{\lambda}(\mathbf{z}^{t+1} - \mathbf{z}^t), \mathbf{z} - \mathbf{z}^{t+1} \rangle \geq 0. \tag{46}$$

Then, we have

$$\langle D_{\mathbf{z}}^t, \mathbf{z} - \mathbf{z}^{t+1} \rangle \geq \frac{1}{\lambda} \langle \mathbf{z}^{t+1} - \mathbf{z}^t, \mathbf{z}^{t+1} - \mathbf{z} \rangle = \frac{1}{\lambda} \|\mathbf{z}^{t+1} - \mathbf{z}^t\|^2 + \frac{1}{\lambda} \langle \mathbf{z}^{t+1} - \mathbf{z}^t, \mathbf{z}^t - \mathbf{z} \rangle \tag{47}$$

Using the above inequalities, we have

$$R(\mathbf{x}^t, \mathbf{y}^t, \mathbf{z})$$

$$\geq R(\mathbf{x}^t, \mathbf{y}^t, \mathbf{z}^{t+1}) + \frac{1}{\lambda} \|\mathbf{z}^{t+1} - \mathbf{z}^t\|^2 + \frac{1}{\lambda} \langle \mathbf{z}^{t+1} - \mathbf{z}^t, \mathbf{z}^t - \mathbf{z} \rangle$$

$$+ \langle \nabla_{\mathbf{z}} R(\mathbf{x}^t, \mathbf{y}^t, \mathbf{z}^t) - D_{\mathbf{z}}^t, \mathbf{z} - \mathbf{z}^{t+1} \rangle + \frac{\mu_g}{2} \|\mathbf{z} - \mathbf{z}^t\|^2 - \frac{L_2^g}{2} \|\mathbf{z}^{t+1} - \mathbf{z}^t\|^2 \tag{48}$$

Let $\mathbf{z} = \mathbf{z}_t^* = \mathbf{z}^*(\mathbf{x}^t, \mathbf{y}^t) = \left[\nabla_{22}^2 g_{\eta_2 \mu_2}(\mathbf{x}^t, \mathbf{y}^t)\right]^{-1} \nabla_2 f_{\eta_1 \mu_1}(\mathbf{x}^t, \mathbf{y}^t)$, then we have

$$R(\mathbf{x}^t, \mathbf{y}^t, \mathbf{z}_t^*)$$

$$\geq R(\mathbf{x}^t, \mathbf{y}^t, \mathbf{z}^{t+1}) + \frac{1}{\lambda} \|\mathbf{z}^{t+1} - \mathbf{z}^t\|^2 + \frac{1}{\lambda} \langle \mathbf{z}^{t+1} - \mathbf{z}^t, \mathbf{z}^t - \mathbf{z}_t^* \rangle$$

$$+ \langle \nabla_{\mathbf{z}} R(\mathbf{x}^t, \mathbf{y}^t, \mathbf{z}^t) - D_{\mathbf{z}}^t, \mathbf{z}_t^* - \mathbf{z}^{t+1} \rangle + \frac{\mu_g}{2} \|\mathbf{z}_t^* - \mathbf{z}^t\|^2 - \frac{L_2^g}{2} \|\mathbf{z}^{t+1} - \mathbf{z}^t\|^2 \tag{49}$$

Due to the strongly convexity of $R(\mathbf{x}^t, \mathbf{y}^t, \mathbf{z})$, we have $R(\mathbf{x}^t, \mathbf{y}^t, \mathbf{z}_t^*) \leq R(\mathbf{x}^t, \mathbf{y}^t, \mathbf{z}^{t+1})$. Thus, we can obtain

$$0 \geq \left(\frac{1}{\lambda} - \frac{L_2^g}{2}\right) \|\mathbf{z}^{t+1} - \mathbf{z}^t\|^2 + \frac{1}{\lambda} \langle \mathbf{z}^{t+1} - \mathbf{z}^t, \mathbf{z}^t - \mathbf{z}_t^* \rangle + \langle \nabla_{\mathbf{z}} R(\mathbf{x}^t, \mathbf{y}^t, \mathbf{z}^t) - D_{\mathbf{z}}^t, \mathbf{z}_t^* - \mathbf{z}^{t+1} \rangle$$

$$+ \frac{\mu_g}{2} \|\mathbf{z}_t^* - \mathbf{z}^t\|^2 \tag{50}$$

For $\langle \mathbf{z}^{t+1} - \mathbf{z}^t, \mathbf{z}^t - \mathbf{z}_t^* \rangle$, we have

$$\langle \mathbf{z}^{t+1} - \mathbf{z}^t, \mathbf{z}^t - \mathbf{z}_t^* \rangle = \frac{1}{2} \|\mathbf{z}^{t+1} - \mathbf{z}_t^*\|^2 - \frac{1}{2} \|\mathbf{z}^t - \mathbf{z}_t^*\|^2 - \frac{1}{2} \|\mathbf{z}^{t+1} - \mathbf{z}^t\|^2 \tag{51}$$

For $\langle \nabla_{\mathbf{z}} R(\mathbf{x}^t, \mathbf{y}^t, \mathbf{z}^t) - D_{\mathbf{z}}^t, \mathbf{z}_t^* - \mathbf{z}^{t+1} \rangle$, we have

$$\langle \nabla_{\mathbf{z}} R(\mathbf{x}^t, \mathbf{y}^t, \mathbf{z}^t) - D_{\mathbf{z}}^t, \mathbf{z}_t^* - \mathbf{z}^{t+1} \rangle$$

$$
\begin{aligned}
&= \langle \nabla_{\mathbf{z}} R(\mathbf{x}^t, \mathbf{y}^t, \mathbf{z}^t) - D_{\mathbf{z}}^t, \mathbf{z}_t^* - \mathbf{z}^t \rangle + \langle \nabla_{\mathbf{z}} R(\mathbf{x}^t, \mathbf{y}^t, \mathbf{z}^t) - D_{\mathbf{z}}^t, \mathbf{z}^t - \mathbf{z}^{t+1} \rangle \\
&\geq -\frac{2}{\mu_g} \|\nabla_{\mathbf{z}} R(\mathbf{x}^t, \mathbf{y}^t, \mathbf{z}^t) - D_{\mathbf{z}}^t\|^2 - \frac{\mu_g}{4} \|\mathbf{z}_t^* - \mathbf{z}^t\|^2 - \frac{\mu_g}{4} \|\mathbf{z}^t - \mathbf{z}^{t+1}\|^2 \\
&\geq -\frac{2}{\mu_g} \|\nabla_{\mathbf{z}} R(\mathbf{x}^t, \mathbf{y}^t, \mathbf{z}^t) - \frac{1}{BB'} \sum_{i=1}^{B} \sum_{j=1}^{B'} \nabla_{\mathbf{z}} R(\mathbf{x}^t, \mathbf{y}^t, \mathbf{z}^t; \bar{\xi}_i^t, \bar{\mathbf{u}}_j^t, \bar{\mathbf{v}}_j^t)\|^2 - \frac{\mu_g}{4} \|\mathbf{z}_t^* - \mathbf{z}^t\|^2 \\
&\quad - \frac{\mu_g}{4} \|\mathbf{z}^t - \mathbf{z}^{t+1}\|^2
\end{aligned}
\tag{52}
$$

Taking expectations on both sides, we have

$$
\begin{aligned}
&\mathbb{E} \langle \nabla_{\mathbf{z}} R(\mathbf{x}^t, \mathbf{y}^t, \mathbf{z}^t) - D_{\mathbf{z}}^t, \mathbf{z}_t^* - \mathbf{z}^{t+1} \rangle \\
&\geq -\frac{2}{\mu_g} \mathbb{E} \|\nabla_{\mathbf{z}} R(\mathbf{x}^t, \mathbf{y}^t, \mathbf{z}^t) - \frac{1}{BB'} \sum_{i=1}^{B} \sum_{j=1}^{B'} \nabla_{\mathbf{z}} R(\mathbf{x}^t, \mathbf{y}^t, \mathbf{z}^t; \bar{\xi}_i^t, \bar{\mathbf{u}}_j^t, \bar{\mathbf{v}}_j^t)\|^2 - \frac{\mu_g}{4} \mathbb{E} \|\mathbf{z}_t^* - \mathbf{z}^t\|^2 \\
&\quad - \frac{\mu_g}{4} \mathbb{E} \|\mathbf{z}^t - \mathbf{z}^{t+1}\|^2 \\
&= -\frac{2\sigma_{\mathbf{z}}^2}{\mu_g BB'} - \frac{\mu_g}{4} \mathbb{E}[\|\mathbf{z}_t^* - \mathbf{z}^t\|^2] - \frac{\mu_g}{4} \mathbb{E}[\|\mathbf{z}^t - \mathbf{z}^{t+1}\|^2]
\end{aligned}
\tag{53}
$$

Using the above inequalities, we have

$$
\begin{aligned}
&\frac{1}{2\lambda} \mathbb{E}[\|\mathbf{z}^{t+1} - \mathbf{z}_t^*\|^2] \\
&\leq \left( \frac{L_2^g}{2} + \frac{\mu_g}{4} - \frac{1}{2\lambda} \right) \mathbb{E}[\|\mathbf{z}^{t+1} - \mathbf{z}^t\|^2] + \left( \frac{1}{2\lambda} - \frac{\mu_g}{4} \right) \mathbb{E}[\|\mathbf{z}^t - \mathbf{z}_t^*\|^2] + \frac{2\sigma_{\mathbf{z}}^2}{\mu_g BB'} \\
&\leq \left( \frac{3L_2^g}{4} - \frac{1}{2\lambda} \right) \mathbb{E}[\|\mathbf{z}^{t+1} - \mathbf{z}^t\|^2] + \left( \frac{1}{2\lambda} - \frac{\mu_g}{4} \right) \mathbb{E}[\|\mathbf{z}^t - \mathbf{z}_t^*\|^2] + \frac{2\sigma_{\mathbf{z}}^2}{\mu_g BB'} \\
&= \left( \frac{1}{2\lambda} - \frac{\mu_g}{4} \right) \mathbb{E}[\|\mathbf{z}^t - \mathbf{z}_t^*\|^2] - \left( \frac{3}{8\lambda} + \frac{1}{8\lambda} - \frac{3L_2^g}{4} \right) \mathbb{E}[\|\mathbf{z}^{t+1} - \mathbf{z}^t\|^2] + \frac{2\sigma_{\mathbf{z}}^2}{\mu_g BB'} \\
&\leq \left( \frac{1}{2\lambda} - \frac{\mu_g}{4} \right) \mathbb{E}[\|\mathbf{z}^t - \mathbf{z}_t^*\|^2] - \frac{3}{8\lambda} \mathbb{E}[\|\mathbf{z}^{t+1} - \mathbf{z}^t\|^2] + \frac{2\sigma_{\mathbf{z}}^2}{\mu_g BB'}
\end{aligned}
\tag{54}
$$

where the second inequality holds by $L_2^g \geq \mu_g$, and the last inequality is due to $0 < \lambda \leq \frac{1}{6L_2^g}$. Therefore, we have

$$
\mathbb{E}[\|\mathbf{z}^{t+1} - \mathbf{z}_t^*\|^2] \leq \left( 1 - \frac{\mu_g \lambda}{2} \right) \mathbb{E}[\|\mathbf{z}^t - \mathbf{z}_t^*\|^2] - \frac{3}{4} \mathbb{E}[\|\mathbf{z}^{t+1} - \mathbf{z}^t\|^2] + \frac{4\lambda \sigma_{\mathbf{z}}^2}{\mu_g BB'}
\tag{55}
$$

For the second term in Eqn. (42), since $\mathbf{z}_{t+1}^* = \mathbf{z}^*(\mathbf{x}^{t+1}, \mathbf{y}^{t+1}) = [\nabla_{22}^2 g_{\eta_2 \mu_2}(\mathbf{x}^{t+1}, \mathbf{y}^{t+1})]^{-1} \nabla_2 f_{\eta_1 \mu_1}(\mathbf{x}^{t+1}, \mathbf{y}^{t+1})$ and $\mathbf{z}_t^* = \mathbf{z}^*(\mathbf{x}^t, \mathbf{y}^t) = [\nabla_{22}^2 g_{\eta_2 \mu_2}(\mathbf{x}^t, \mathbf{y}^t)]^{-1} \nabla_2 f_{\eta_1 \mu_1}(\mathbf{x}^t, \mathbf{y}^t)$, we have

$$
\begin{aligned}
&\|\mathbf{z}_t^* - \mathbf{z}_{t+1}^*\|^2 \\
&= \left\| [\nabla_{22}^2 g_{\eta_2 \mu_2}(\mathbf{x}^{t+1}, \mathbf{y}^{t+1})]^{-1} \nabla_2 f_{\eta_1 \mu_1}(\mathbf{x}^{t+1}, \mathbf{y}^{t+1}) - [\nabla_{22}^2 g_{\eta_2 \mu_2}(\mathbf{x}^t, \mathbf{y}^t)]^{-1} \nabla_2 f_{\eta_1 \mu_1}(\mathbf{x}^t, \mathbf{y}^t) \right\|^2 \\
&\leq 2 \left\| [\nabla_{22}^2 g_{\eta_2 \mu_2}(\mathbf{x}^{t+1}, \mathbf{y}^{t+1})]^{-1} (\nabla_2 f_{\eta_1 \mu_1}(\mathbf{x}^{t+1}, \mathbf{y}^{t+1}) - \nabla_2 f_{\eta_1 \mu_1}(\mathbf{x}^t, \mathbf{y}^t)) \right\|^2 \\
&\quad + 2 \left\| ([\nabla_{22}^2 g_{\eta_2 \mu_2}(\mathbf{x}^{t+1}, \mathbf{y}^{t+1})]^{-1} - [\nabla_{22}^2 g_{\eta_2 \mu_2}(\mathbf{x}^t, \mathbf{y}^t)]^{-1}) \nabla_2 f_{\eta_1 \mu_1}(\mathbf{x}^t, \mathbf{y}^t)) \right\|^2 \\
&\leq \frac{4L_1^f}{\mu_g} (\|\mathbf{x}^{t+1} - \mathbf{x}^t\|^2 + \|\mathbf{y}^{t+1} - \mathbf{y}^t\|^2) + \frac{4L_0^f L_2^g}{\mu_g^2} (\|\mathbf{x}^{t+1} - \mathbf{x}^t\|^2 + \|\mathbf{y}^{t+1} - \mathbf{y}^t\|^2) \\
&= \left( \frac{4L_1^f}{\mu_g} + \frac{4L_0^f L_2^g}{\mu_g^2} \right) (\|\mathbf{x}^{t+1} - \mathbf{x}^t\|^2 + \|\mathbf{y}^{t+1} - \mathbf{y}^t\|^2),
\end{aligned}
\tag{56}
$$

where the second inequality holds by Lemma 7.

Combining Eqn. (55), Eqn. (56) and Eqn. (42), we have

$$\mathbb{E}\|\mathbf{z}^{t+1} - \mathbf{z}^*_{t+1}\|^2$$

$$\leq (1 + \frac{\mu_g \lambda}{4}) \mathbb{E}\|\mathbf{z}^{t+1} - \mathbf{z}^*_t\|^2 + (1 + \frac{4}{\mu \lambda}) \mathbb{E}\|\mathbf{z}^*_t - \mathbf{z}^*_{t+1}\|^2$$

$$\leq \left(1 + \frac{\mu_g \lambda}{4}\right) \left(1 - \frac{\mu_g \lambda}{2}\right) \mathbb{E}[\|\mathbf{z}^t - \mathbf{z}^*_t\|^2] - \left(1 + \frac{\mu_g \lambda}{4}\right) \frac{3}{4} \mathbb{E}[\|\mathbf{z}^{t+1} - \mathbf{z}^t\|^2]$$

$$+ \left(1 + \frac{\mu_g \lambda}{4}\right) \frac{4\lambda \sigma_{\mathbf{z}}^2}{\mu_g BB'} + \left(1 + \frac{4}{\mu \lambda}\right) \left(\frac{4L_1^f}{\mu_g} + \frac{4L_0^f L_2^g}{\mu_g^2}\right) (\mathbb{E}[\|\mathbf{x}^{t+1} - \mathbf{x}^t\|^2] + \mathbb{E}[\|\mathbf{y}^{t+1} - \mathbf{y}^t\|^2])$$

$$(57)$$

Since $0 < \lambda \leq \frac{1}{6L_2^g}$ and $L_2^g \geq \mu_g$, we have $\lambda \leq \frac{1}{6L_2^g} \leq \frac{1}{6\mu_g}$. Then, we have

$$\left(1 + \frac{\mu_g \lambda}{4}\right) \left(1 - \frac{\mu_g \lambda}{2}\right) = 1 - \frac{\mu_g \lambda}{2} + \frac{\mu_g \lambda}{4} - \frac{\mu^2 \lambda^2}{8} \leq 1 - \frac{\mu_g \lambda}{4}, \tag{58}$$

$$-\left(1 + \frac{\mu_g \lambda}{4}\right) \frac{3}{4} \leq -\frac{3}{4}, \tag{59}$$

$$\left(1 + \frac{\mu_g \lambda}{4}\right) \frac{4\lambda \sigma_{\mathbf{z}}^2}{\mu_g BB'} \leq \left(1 + \frac{1}{24}\right) \frac{4\lambda \sigma_{\mathbf{z}}^2}{\mu_g BB'} = \frac{25\lambda \sigma_{\mathbf{z}}^2}{6\mu_g BB'}, \tag{60}$$

$$\left(1 + \frac{4}{\mu_g \lambda}\right) \leq \frac{5}{3}. \tag{61}$$

Therefore, we have

$$\mathbb{E}\|\mathbf{z}^{t+1} - \mathbf{z}^*_{t+1}\|^2$$

$$\leq \left(1 - \frac{\mu_g \lambda}{4}\right) \mathbb{E}[\|\mathbf{z}^t - \mathbf{z}^*_t\|^2] - \frac{3}{4} \mathbb{E}[\|\mathbf{z}^{t+1} - \mathbf{z}^t\|^2]$$

$$+ \frac{25\lambda \sigma_{\mathbf{z}}^2}{6\mu_g BB'} + \frac{20}{3} \left(\frac{L_1^f}{\mu_g} + \frac{L_0^f L_2^g}{\mu_g^2}\right) (\mathbb{E}[\|\mathbf{x}^{t+1} - \mathbf{x}^t\|^2] + \mathbb{E}[\|\mathbf{y}^{t+1} - \mathbf{y}^t\|^2]) \tag{62}$$

$$\square$$

## D.2 DETAILED PROOF OF LEMMA 5

*Proof.* In our algorithm, we have

$$\mathbf{x}^{t+1} = \arg\min_{\mathbf{x} \in \mathbb{R}^{d_1}} \left\{ \langle D_{\mathbf{x}}^t, \mathbf{x} \rangle + h(\mathbf{x}) + \mathcal{B}_{\psi_t}(\mathbf{x}, \mathbf{x}^t) \right\} \tag{63}$$

where $\mathcal{B}_{\psi_t}(\mathbf{x}_1, \mathbf{x}_2) = \psi(\mathbf{x}_1) - \psi(\mathbf{x}_2) - \langle \nabla \psi(\mathbf{x}_2), \mathbf{x}_1 - \mathbf{x}_2 \rangle$. Since $\mathcal{G}^t = \frac{1}{\alpha}(\mathbf{x}^{t+1} - \mathbf{x}^t)$, we have for all $t \geq 1$,

$$\langle D_{\mathbf{x}}^t, \mathcal{G}^t \rangle \geq \rho \|\mathcal{G}^t\|^2 + \frac{1}{\alpha}(h(\mathbf{x}^{t+1}) - h(\mathbf{x}^t)) \tag{64}$$

where $\rho > 0$ depends on $\rho$-strongly convex function $\psi_t(\mathbf{x})$.

According to Lemma 8, using the above result, we have

$$F(\mathbf{x}^{t+1})$$

$$\leq F(\mathbf{x}^t) + \langle \nabla F(\mathbf{x}^t), \mathbf{x}^{t+1} - \mathbf{x}^t \rangle + \frac{L_F}{2} \|\mathbf{x}^{t+1} - \mathbf{x}^t\|^2$$

$$= F(\mathbf{x}^t) - \alpha \langle \nabla F(\mathbf{x}^t), \mathcal{G}^t \rangle + \frac{\alpha^2 L_F}{2} \|\mathcal{G}^t\|^2$$

$$= F(\mathbf{x}^t) - \alpha \langle D_{\mathbf{x}}^t, \mathcal{G}^t \rangle + \alpha \langle D_{\mathbf{x}}^t - \nabla F(\mathbf{x}^t), \mathcal{G}^t \rangle + \frac{\alpha^2 L_F}{2} \|\mathcal{G}^t\|^2$$

$$\leq F(\mathbf{x}^t) - \alpha\rho\|\mathcal{G}^t\|^2 - h(\mathbf{x}^{t+1}) + h(\mathbf{x}^t) + \alpha\langle D_{\mathbf{x}}^t - \nabla F(\mathbf{x}^t), \mathcal{G}^t\rangle + \frac{\alpha^2 L_F}{2}\|\mathcal{G}^t\|^2$$

$$\leq F(\mathbf{x}^t) - \alpha\rho\|\mathcal{G}^t\|^2 - h(\mathbf{x}^{t+1}) + h(\mathbf{x}^t) + \alpha\|D_{\mathbf{x}}^t - \nabla F(\mathbf{x}^t)\|\|\mathcal{G}^t\| + \frac{\alpha^2 L_F}{2}\|\mathcal{G}^t\|^2$$

$$\leq F(\mathbf{x}^t) - \alpha\rho\|\mathcal{G}^t\|^2 - h(\mathbf{x}^{t+1}) + h(\mathbf{x}^t) + \frac{\alpha}{\rho}\|D_{\mathbf{x}}^t - \nabla F(\mathbf{x}^t)\|^2 + \frac{\alpha\rho}{4}\|\mathcal{G}^t\|^2 + \frac{\alpha^2 L_F}{2}\|\mathcal{G}^t\|^2$$

$$\leq F(\mathbf{x}^t) + (\frac{\alpha^2 L_F}{2} - \frac{3\alpha\rho}{4})\|\mathcal{G}^t\|^2 - h(\mathbf{x}^{t+1}) + h(\mathbf{x}^t) + \frac{\alpha}{\rho}\|D_{\mathbf{x}}^t - \nabla F(\mathbf{x}^t)\|^2$$

$$\leq F(\mathbf{x}^t) + (\frac{\alpha^2 L_F}{2} - \frac{3\alpha\rho}{4})\|\mathcal{G}^t\|^2 - h(\mathbf{x}^{t+1}) + h(\mathbf{x}^t) + \frac{2\alpha}{\rho}\|D_{\mathbf{x}}^t - \nabla F_{\eta\mu}(\mathbf{x}^t)\|^2$$

$$+ \frac{2\alpha}{\rho}\|\nabla F_{\eta\mu}(\mathbf{x}^t) - \nabla F(\mathbf{x}^t)\|^2$$

$$\leq F(\mathbf{x}^t) + (\frac{\alpha^2 L_F}{2} - \frac{3\alpha\rho}{4})\|\mathcal{G}^t\|^2 - h(\mathbf{x}^{t+1}) + h(\mathbf{x}^t) + \frac{2\alpha}{\rho}\|D_{\mathbf{x}}^t - \nabla F_{\eta\mu}(\mathbf{x}^t)\|^2 + \frac{2\alpha}{\rho}A$$

$$(65)$$

where the last inequality is due to Proposition 3.1 in (Aghasi & Ghadimi, 2024) and $\sqrt{A} = L_1^f\sqrt{\frac{2L_1^g}{\mu_g}(\eta_2^2 d_1 + \mu_2^2 d_2)} + \frac{L_1^f}{2}(\eta_1(d_1 + 3)^{\frac{3}{2}} + \frac{\mu_1^2}{\eta_1}d_2 d_1^{\frac{1}{2}} + \frac{\eta_1^2}{\mu_1}d_1 d_2^{\frac{1}{2}} + \mu_1(d_2 + 3)^{\frac{3}{2}})$.

For $\|D_{\mathbf{x}}^t - \nabla F_{\eta\mu}(\mathbf{x}^t)\|^2$, since $D_{\mathbf{x}}^t = \frac{1}{BB'}\sum_{i=1}^B\sum_{j=1}^{B'}\bar{\nabla}f_{\eta\mu}(\mathbf{x}^t, \mathbf{y}^t, \mathbf{z}^t; \bar{\xi}_i^t, \bar{\mathbf{u}}_j^t, \bar{\mathbf{v}}_j^t) = \hat{\nabla}_1 f_{\eta_1\mu_1}(\mathbf{x}^t, \mathbf{y}^t; \xi_i^t, \mathbf{u}_{1,j}^t, \mathbf{v}_{1,j}^t) - \hat{\nabla}_{12}^2 g_{\eta_2\mu_2}(\mathbf{x}^t, \mathbf{y}^t; \zeta_i^t, \mathbf{u}_{2,j}^t, \mathbf{v}_{2,j}^t)\mathbf{z}^t$, $\bar{\nabla}f_{\eta\mu}(\mathbf{x}^t, \mathbf{y}^t, \mathbf{z}^t) = \nabla_1 f_{\eta_1\mu_1}(\mathbf{x}^t, \mathbf{y}^t) - \nabla_{12}^2 g_{\eta_2\mu_2}(\mathbf{x}^t, \mathbf{y}^t)\mathbf{z}^t$ and $\nabla F_{\eta\mu}(\mathbf{x}^t) = \nabla_1 f_{\eta_1\mu_1}(\mathbf{x}^t, \mathbf{y}_{\eta_2\mu_2}^*(\mathbf{x}^t)) - \nabla_{12}^2 g_{\eta_2\mu_2}(\mathbf{x}^t, \mathbf{y}_{\eta_2\mu_2}^*(\mathbf{x}^t))\mathbf{z}^*(\mathbf{x}^t)$, we have

$$\|D_{\mathbf{x}}^t - \nabla F_{\eta\mu}(\mathbf{x}^t)\|^2$$

$$= 2\|\frac{1}{BB'}\sum_{i=1}^B\sum_{j=1}^{B'}\bar{\nabla}f_{\eta\mu}(\mathbf{x}^t, \mathbf{y}^t, \mathbf{z}^t; \bar{\xi}_i^t, \bar{\mathbf{u}}_j^t, \bar{\mathbf{v}}_j^t) - \bar{\nabla}f_{\eta\mu}(\mathbf{x}^t, \mathbf{y}^t, \mathbf{z}^t)\|^2$$

$$+ 2\|\bar{\nabla}f_{\eta\mu}(\mathbf{x}^t, \mathbf{y}^t, \mathbf{z}^t) - \nabla F_{\eta\mu}(\mathbf{x}^t)\|^2$$

$$= 2\|\frac{1}{BB'}\sum_{i=1}^B\sum_{j=1}^{B'}\bar{\nabla}f_{\eta\mu}(\mathbf{x}^t, \mathbf{y}^t, \mathbf{z}^t; \bar{\xi}_i^t, \bar{\mathbf{u}}_j^t, \bar{\mathbf{v}}_j^t) - \bar{\nabla}f_{\eta\mu}(\mathbf{x}^t, \mathbf{y}^t, \mathbf{z}^t)\|^2$$

$$+ 2\|\nabla_1 f_{\eta_1\mu_1}(\mathbf{x}^t, \mathbf{y}^t) - \nabla_1 f_{\eta_1\mu_1}(\mathbf{x}^t, \mathbf{y}_{\eta_2\mu_2}^*(\mathbf{x}^t))$$

$$+ \nabla_{12}^2 g_{\eta_2\mu_2}(\mathbf{x}^t, \mathbf{y}^t)\mathbf{z}^t - \nabla_{12}^2 g_{\eta_2\mu_2}(\mathbf{x}^t, \mathbf{y}_{\eta_2\mu_2}^*(\mathbf{x}^t))\mathbf{z}^t$$

$$+ \nabla_{12}^2 g_{\eta_2\mu_2}(\mathbf{x}^t, \mathbf{y}_{\eta_2\mu_2}^*(\mathbf{x}^t))\mathbf{z}^t - \nabla_{12}^2 g_{\eta_2\mu_2}(\mathbf{x}^t, \mathbf{y}_{\eta_2\mu_2}^*(\mathbf{x}^t))\mathbf{z}^*(\mathbf{x}^t, \mathbf{y}^t)$$

$$+ \nabla_{12}^2 g_{\eta_2\mu_2}(\mathbf{x}^t, \mathbf{y}_{\eta_2\mu_2}^*(\mathbf{x}^t))\mathbf{z}^*(\mathbf{x}^t, \mathbf{y}^t) - \nabla_{12}^2 g_{\eta_2\mu_2}(\mathbf{x}^t, \mathbf{y}_{\eta_2\mu_2}^*(\mathbf{x}^t))\mathbf{z}^*(\mathbf{x}^t)\|^2$$

$$\leq 2\|\frac{1}{BB'}\sum_{i=1}^B\sum_{j=1}^{B'}\bar{\nabla}f_{\eta\mu}(\mathbf{x}^t, \mathbf{y}^t, \mathbf{z}^t; \bar{\xi}_i^t, \bar{\mathbf{u}}_j^t, \bar{\mathbf{v}}_j^t) - \bar{\nabla}f_{\eta\mu}(\mathbf{x}^t, \mathbf{y}^t, \mathbf{z}^t)\|^2$$

$$+ 8\|\nabla_1 f_{\eta_1\mu_1}(\mathbf{x}^t, \mathbf{y}^t) - \nabla_1 f_{\eta_1\mu_1}(\mathbf{x}^t, \mathbf{y}_{\eta_2\mu_2}^*(\mathbf{x}^t))\|^2$$

$$+ 8\left\|\left(\nabla_{12}^2 g_{\eta_2\mu_2}(\mathbf{x}^t, \mathbf{y}^t) - \nabla_{12}^2 g_{\eta_2\mu_2}(\mathbf{x}^t, \mathbf{y}_{\eta_2\mu_2}^*(\mathbf{x}^t))\right)\mathbf{z}^t\right\|^2$$

$$+ 8\left\|\nabla_{12}^2 g_{\eta_2\mu_2}(\mathbf{x}^t, \mathbf{y}_{\eta_2\mu_2}^*(\mathbf{x}^t))(\mathbf{z}^t - \mathbf{z}^*(\mathbf{x}^t, \mathbf{y}^t))\right\|^2$$

$$+ 8\|\nabla_{12}^2 g_{\eta_2\mu_2}(\mathbf{x}^t, \mathbf{y}_{\eta_2\mu_2}^*(\mathbf{x}^t))(\mathbf{z}^*(\mathbf{x}^t, \mathbf{y}^t) - \mathbf{z}^*(\mathbf{x}^t))\|^2$$

$$\leq 2\|\frac{1}{BB'}\sum_{i=1}^B\sum_{j=1}^{B'}\bar{\nabla}f_{\eta\mu}(\mathbf{x}^t, \mathbf{y}^t, \mathbf{z}^t; \bar{\xi}_i^t, \bar{\mathbf{u}}_j^t, \bar{\mathbf{v}}_j^t) - \bar{\nabla}f_{\eta\mu}(\mathbf{x}^t, \mathbf{y}^t, \mathbf{z}^t)\|^2$$

$$+ (8(L_1^f)^2 + 8(L_2^g r_{\mathbf{z}})^2)\|\mathbf{y}^t - \mathbf{y}_{\eta_2\mu_2}^*(\mathbf{x}^t)\|^2$$

$$+ 8(L_1^g)^2\left\|\mathbf{z}^t - \mathbf{z}_t^*\right\|^2$$

$$+ 8(L_1^g)^2 \|\mathbf{z}^*(\mathbf{x}^t, \mathbf{y}^t) - \mathbf{z}^*(\mathbf{x}^t)\|^2$$

$$(66)$$

For the last term, we have

$$\|\mathbf{z}^*(\mathbf{x}^t, \mathbf{y}^t) - \mathbf{z}^*(\mathbf{x}^t)\|^2$$

$$= \| \left[ \nabla_{22}^2 g_{\eta_2 \mu_2}(\mathbf{x}^t, \mathbf{y}^t) \right]^{-1} \nabla_\mathbf{y} f_{\eta_1 \mu_1}(\mathbf{x}^t, \mathbf{y}^t) - \left[ \nabla_{22}^2 g_{\eta_2 \mu_2}(\mathbf{x}^t, \mathbf{y}_{\eta_2 \mu_2}^*(\mathbf{x}^t)) \right]^{-1} \nabla_\mathbf{y} f_{\eta_1 \mu_1}(\mathbf{x}^t, \mathbf{y}_{\eta_2 \mu_2}^*(\mathbf{x}^t)) \|^2$$

$$\leq 2\| \left[ \nabla_{22}^2 g_{\eta_2 \mu_2}(\mathbf{x}^t, \mathbf{y}^t) \right]^{-1} \left( \nabla_\mathbf{y} f_{\eta_1 \mu_1}(\mathbf{x}^t, \mathbf{y}^t) - \nabla_\mathbf{y} f_{\eta_1 \mu_1}(\mathbf{x}^t, \mathbf{y}_{\eta_2 \mu_2}^*(\mathbf{x}^t))) \|^2$$

$$+ 2 \left\| \left( \left[ \nabla_{22}^2 g_{\eta_2 \mu_2}(\mathbf{x}^t, \mathbf{y}^t) \right]^{-1} - \left[ \nabla_{22}^2 g_{\eta_2 \mu_2}(\mathbf{x}^t, \mathbf{y}_{\eta_2 \mu_2}^*(\mathbf{x}^t)) \right]^{-1} \right) \nabla_\mathbf{y} f_{\eta_1 \mu_1}(\mathbf{x}^t, \mathbf{y}_{\eta_2 \mu_2}^*(\mathbf{x}^t)) \right\|^2$$

$$\leq \frac{2(L_1^f)^2}{\mu_g^2} \|\mathbf{y}^t - \mathbf{y}_{\eta_2 \mu_2}^*(\mathbf{x}^t))\|^2 + \frac{2(L_0^f L_2^g)^2}{\mu_g^4} \|\mathbf{y}^t - \mathbf{y}_{\eta_2 \mu_2}^*(\mathbf{x}^t))\|^2$$

$$\leq \left( \frac{2(L_1^f)^2}{\mu_g^2} + \frac{2(L_0^f L_2^g)^2}{\mu_g^4} \right) \|\mathbf{y}^t - \mathbf{y}_{\eta_2 \mu_2}^*(\mathbf{x}^t))\|^2 \qquad (67)$$

Combining Eqn (70), Eqn (66) and Eqn (67), we can obtain

$$F(\mathbf{x}^{t+1})$$

$$\leq F(\mathbf{x}^t) + (\frac{\alpha^2 L_F}{2} - \frac{3\alpha\rho}{4})\|\mathcal{G}^t\|^2 - h(\mathbf{x}^{t+1}) + h(\mathbf{x}^t)$$

$$+ \frac{2\alpha}{\rho}(2\| \frac{1}{BB'} \sum_{i=1}^B \sum_{j=1}^{B'} \bar{\nabla} f_{\eta\mu}(\mathbf{x}^t, \mathbf{y}^t, \mathbf{z}^t; \bar{\xi}_i^t, \bar{\mathbf{u}}_j^t, \bar{\mathbf{v}}_j^t) - \bar{\nabla} f_{\eta\mu}(\mathbf{x}^t, \mathbf{y}^t, \mathbf{z}^t)\|^2$$

$$+ (8(L_1^f)^2 + 8(L_2^g r_\mathbf{z})^2)\|\mathbf{y}^t - \mathbf{y}_{\eta_2 \mu_2}^*(\mathbf{x}^t)\|^2$$

$$+ 8(L_1^g)^2 \left\| \mathbf{z}^t - \mathbf{z}_t^* \right\|^2 + 8(L_1^g)^2 \left( \frac{2(L_1^f)^2}{\mu_g^2} + \frac{2(L_0^f L_2^g)^2}{\mu_g^4} \right) \|\mathbf{y}^t - \mathbf{y}_{\eta_2 \mu_2}^*(\mathbf{x}^t))\|^2) + \frac{2\alpha}{\rho} A$$

$$= F(\mathbf{x}^t) + (\frac{\alpha^2 L_F}{2} - \frac{3\alpha\rho}{4})\|\mathcal{G}^t\|^2 - h(\mathbf{x}^{t+1}) + h(\mathbf{x}^t)$$

$$+ \frac{4\alpha}{\rho}\| \frac{1}{BB'} \sum_{i=1}^B \sum_{j=1}^{B'} \bar{\nabla} f_{\eta\mu}(\mathbf{x}^t, \mathbf{y}^t, \mathbf{z}^t; \bar{\xi}_i^t, \bar{\mathbf{u}}_j^t, \bar{\mathbf{v}}_j^t) - \bar{\nabla} f_{\eta\mu}(\mathbf{x}^t, \mathbf{y}^t, \mathbf{z}^t)\|^2$$

$$+ \frac{2\alpha}{\rho} \left( 8(L_1^f)^2 + 8(L_2^g r_\mathbf{z})^2 + 8(L_1^g)^2 \left( \frac{2(L_1^f)^2}{\mu_g^2} + \frac{2(L_0^f L_2^g)^2}{\mu_g^4} \right) \right) \|\mathbf{y}^t - \mathbf{y}_{\eta_2 \mu_2}^*(\mathbf{x}^t)\|^2$$

$$+ \frac{16\alpha(L_1^g)^2}{\rho} \left\| \mathbf{z}^t - \mathbf{z}_t^* \right\|^2 + \frac{2\alpha}{\rho} A \qquad (68)$$

Then, taking expectation on both sides, we have

$$F(\mathbf{x}^{t+1})$$

$$\leq F(\mathbf{x}^t) + (\frac{\alpha^2 L_F}{2} - \frac{3\alpha\rho}{4})\mathbb{E}[\|\mathcal{G}^t\|^2] - h(\mathbf{x}^{t+1}) + h(\mathbf{x}^t)$$

$$+ \frac{2\alpha}{\rho} \left( 8(L_1^f)^2 + 8(L_2^g r_\mathbf{z})^2 + 8(L_1^g)^2 \left( \frac{2(L_1^f)^2}{\mu_g^2} + \frac{2(L_0^f L_2^g)^2}{\mu_g^4} \right) \right) \mathbb{E}[\|\mathbf{y}^t - \mathbf{y}_{\eta_2 \mu_2}^*(\mathbf{x}^t)\|^2]$$

$$+ \frac{16\alpha(L_1^g)^2}{\rho} \mathbb{E}[\left\| \mathbf{z}^t - \mathbf{z}_t^* \right\|^2] + \frac{2\alpha}{\rho} A + \frac{4\alpha \sigma_\mathbf{x}^2}{\rho BB'}$$

$$\leq F(\mathbf{x}^t) - \frac{3\alpha\rho}{8}\mathbb{E}[\|\mathcal{G}^t\|^2] - h(\mathbf{x}^{t+1}) + h(\mathbf{x}^t)$$

$$+ \frac{2\alpha}{\rho} \left( 8(L_1^f)^2 + 8(L_2^g r_\mathbf{z})^2 + 8(L_1^g)^2 \left( \frac{2(L_1^f)^2}{\mu_g^2} + \frac{2(L_0^f L_2^g)^2}{\mu_g^4} \right) \right) \mathbb{E}[\|\mathbf{y}^t - \mathbf{y}_{\eta_2 \mu_2}^*(\mathbf{x}^t)\|^2]$$

$$+ \frac{16\alpha(L_1^g)^2}{\rho}\mathbb{E}[\|\mathbf{z}^t - \mathbf{z}_t^*\|^2] + \frac{2\alpha}{\rho}A + \frac{4\alpha\sigma_{\mathbf{x}}^2}{\rho BB'} \tag{69}$$

where the last inequality is due to $0 < \alpha \leq \frac{3\rho}{4L_F}$. Since $\Phi(\mathbf{x}^t) = F(\mathbf{x}^t) + h(\mathbf{x}^t)$, we have

$$\Phi(\mathbf{x}^{t+1})$$
$$\leq \Phi(\mathbf{x}^t) - \frac{3\alpha\rho}{8}\mathbb{E}[\|\mathcal{G}^t\|^2] + \frac{16\alpha(L_1^g)^2}{\rho}\mathbb{E}[\|\mathbf{z}^t - \mathbf{z}_t^*\|^2] + \frac{2\alpha}{\rho}A + \frac{4\alpha\sigma_{\mathbf{x}}^2}{\rho BB'}$$
$$+ \frac{2\alpha}{\rho}\left(8(L_1^f)^2 + 8(L_2^g r_{\mathbf{z}})^2 + 8(L_1^g)^2\left(\frac{2(L_1^f)^2}{\mu_g^2} + \frac{2(L_0^f L_2^g)^2}{\mu_g^4}\right)\right)\mathbb{E}[\|\mathbf{y}^t - \mathbf{y}_{\eta_2\mu_2}^*(\mathbf{x}^t)\|^2] \tag{70}$$

$\square$

### D.3 DETAILED PROOF OF LEMMA 6

*Proof.* Here, we rewrite the rule $\mathbf{y}^{t+1} = \mathbf{y}^t - \beta\tau D_{\mathbf{y}}^t$ in our algorithm as $\mathbf{y}^{t+1} = \mathbf{y}^t + \tau(\tilde{\mathbf{y}}^{t+1} - \mathbf{y}^t)$ and $\tilde{\mathbf{y}}^{t+1} = \mathbf{y}^t - \beta D_{\mathbf{y}}^t$. Then, using the $\mu_g$-smoothness of $g_{\eta_2\mu_2}$ in Lemma 7, we have

$$g_{\eta_2\mu_2}(\mathbf{x}^t, \mathbf{y})$$
$$\geq g_{\eta_2\mu_2}(\mathbf{x}^t, \mathbf{y}^t) + \langle\nabla_2 g_{\eta_2\mu_2}(\mathbf{x}^t, \mathbf{y}^t), \mathbf{y} - \mathbf{y}^t\rangle + \frac{\mu_g}{2}\|\mathbf{y} - \mathbf{y}^t\|^2$$
$$= g_{\eta_2\mu_2}(\mathbf{x}^t, \mathbf{y}^t) + \langle D_{\mathbf{y}}^t, \mathbf{y} - \tilde{\mathbf{y}}^{t+1}\rangle + \langle\nabla_2 g_{\eta_2\mu_2}(\mathbf{x}^t, \mathbf{y}^t) - D_{\mathbf{y}}^t, \mathbf{y} - \tilde{\mathbf{y}}^{t+1}\rangle$$
$$+ \langle\nabla_2 g_{\eta_2\mu_2}(\mathbf{x}^t, \mathbf{y}^t), \tilde{\mathbf{y}}^{t+1} - \mathbf{y}^t\rangle + \frac{\mu_g}{2}\|\mathbf{y} - \mathbf{y}^t\|^2 \tag{71}$$

Using the $L_2^g$-smoothness of $g_{\eta_2\mu_2}$ in Lemma 7, we have

$$g_{\eta_2\mu_2}(\mathbf{x}^t, \tilde{\mathbf{y}}^{t+1}) \leq g_{\eta_2\mu_2}(\mathbf{x}^t, \mathbf{y}^t) + \langle\nabla_2 g_{\eta_2\mu_2}(\mathbf{x}^t, \mathbf{y}^t), \tilde{\mathbf{y}}^{t+1} - \mathbf{y}^t\rangle + \frac{L_2^g}{2}\|\tilde{\mathbf{y}}^{t+1} - \mathbf{y}^t\|^2. \tag{72}$$

Then, combining the above inequalities, we can obtain

$$g_{\eta_2\mu_2}(\mathbf{x}^t, \mathbf{y}) \geq g_{\eta_2\mu_2}(\mathbf{x}^t, \tilde{\mathbf{y}}^{t+1}) + \langle D_{\mathbf{y}}^t, \mathbf{y} - \tilde{\mathbf{y}}^{t+1}\rangle - \frac{L_2^g}{2}\|\tilde{\mathbf{y}}^{t+1} - \mathbf{y}^t\|^2$$
$$+ \langle\nabla_2 g_{\eta_2\mu_2}(\mathbf{x}^t, \mathbf{y}^t) - D_{\mathbf{y}}^t, \mathbf{y} - \tilde{\mathbf{y}}^{t+1}\rangle + \frac{\mu_g}{2}\|\mathbf{y} - \mathbf{y}^t\|^2 \tag{73}$$

Since $\tilde{\mathbf{y}}^{t+1} = \mathbf{y}^t - \beta D_{\mathbf{y}}^t$, we have

$$\langle D_{\mathbf{y}}^t, \mathbf{y} - \tilde{\mathbf{y}}^{t+1}\rangle$$
$$= \frac{1}{\beta}\langle\tilde{\mathbf{y}}^{t+1} - \mathbf{y}^t, \tilde{\mathbf{y}}^{t+1} - \mathbf{y}\rangle$$
$$= \frac{1}{\beta}\|\tilde{\mathbf{y}}^{t+1} - \mathbf{y}^t\|^2 + \frac{1}{\beta}\langle\tilde{\mathbf{y}}^{t+1} - \mathbf{y}^t, \mathbf{y}^t - \mathbf{y}\rangle. \tag{74}$$

Thus, we can obtain

$$g_{\eta_2\mu_2}(\mathbf{x}^t, \mathbf{y})$$
$$\geq g_{\eta_2\mu_2}(\mathbf{x}^t, \tilde{\mathbf{y}}^{t+1}) + \frac{1}{\beta}\|\tilde{\mathbf{y}}^{t+1} - \mathbf{y}^t\|^2 + \frac{1}{\beta}\langle\tilde{\mathbf{y}}^{t+1} - \mathbf{y}^t, \mathbf{y}^t - \mathbf{y}\rangle - \frac{L_2^g}{2}\|\tilde{\mathbf{y}}^{t+1} - \mathbf{y}^t\|^2$$
$$+ \langle\nabla_2 g_{\eta_2\mu_2}(\mathbf{x}^t, \mathbf{y}^t) - D_{\mathbf{y}}^t, \mathbf{y} - \tilde{\mathbf{y}}^{t+1}\rangle + \frac{\mu_g}{2}\|\mathbf{y} - \mathbf{y}^t\|^2. \tag{75}$$

Then, setting $\mathbf{y} = \mathbf{y}_{\eta_2\mu_2}^*(\mathbf{x}^t)$, we have

$$g_{\eta_2\mu_2}(\mathbf{x}^t, \mathbf{y}_{\eta_2\mu_2}^*(\mathbf{x}^t))$$
$$\geq g_{\eta_2\mu_2}(\mathbf{x}^t, \tilde{\mathbf{y}}^{t+1}) + \frac{1}{\beta}\|\tilde{\mathbf{y}}^{t+1} - \mathbf{y}^t\|^2 + \frac{1}{\beta}\langle\tilde{\mathbf{y}}^{t+1} - \mathbf{y}^t, \mathbf{y}^t - \mathbf{y}_{\eta_2\mu_2}^*(\mathbf{x}^t)\rangle - \frac{L_2^g}{2}\|\tilde{\mathbf{y}}^{t+1} - \mathbf{y}^t\|^2$$

$$+ \langle \nabla_2 g_{\eta_2\mu_2}(\mathbf{x}^t, \mathbf{y}^t) - D_{\mathbf{y}}^t, \mathbf{y}_{\eta_2\mu_2}^*(\mathbf{x}^t) - \tilde{\mathbf{y}}^{t+1} \rangle + \frac{\mu_g}{2} \|\mathbf{y}_{\eta_2\mu_2}^*(\mathbf{x}^t) - \mathbf{y}^t\|^2. \tag{76}$$

Due to the strong convexity of $g_{\eta_2\mu_2}$, we have $g_{\eta_2\mu_2}(\mathbf{x}^t, \mathbf{y}_{\eta_2\mu_2}^*(\mathbf{x}^t)) \leq g_{\eta_2\mu_2}(\mathbf{x}^t, \tilde{\mathbf{y}}^{t+1})$. Therefore, we can obtain

$$0 \geq \left(\frac{1}{\beta} - \frac{L_2^g}{2}\right) \|\tilde{\mathbf{y}}^{t+1} - \mathbf{y}^t\|^2 + \frac{1}{\beta} \langle \tilde{\mathbf{y}}^{t+1} - \mathbf{y}^t, \mathbf{y}^t - \mathbf{y}_{\eta_2\mu_2}^*(\mathbf{x}^t) \rangle$$
$$+ \langle \nabla_2 g_{\eta_2\mu_2}(\mathbf{x}^t, \mathbf{y}^t) - D_{\mathbf{y}}^t, \mathbf{y}_{\eta_2\mu_2}^*(\mathbf{x}^t) - \tilde{\mathbf{y}}^{t+1} \rangle + \frac{\mu_g}{2} \|\mathbf{y}_{\eta_2\mu_2}^*(\mathbf{x}^t) - \mathbf{y}^t\|^2. \tag{77}$$

For $\langle \tilde{\mathbf{y}}^{t+1} - \mathbf{y}^t, \mathbf{y}^t - \mathbf{y}_{\eta_2\mu_2}^*(\mathbf{x}^t) \rangle$, we have

$$\langle \tilde{\mathbf{y}}^{t+1} - \mathbf{y}^t, \mathbf{y}^t - \mathbf{y}_{\eta_2\mu_2}^*(\mathbf{x}^t) \rangle = \frac{1}{2\tau} \|\mathbf{y}^{t+1} - \mathbf{y}_{\eta_2\mu_2}^*(\mathbf{x}^t)\|^2 - \frac{\tau}{2} \|\tilde{\mathbf{y}}^{t+1} - \mathbf{y}^t\|^2 - \frac{1}{2\tau} \|\mathbf{y}^t - \mathbf{y}_{\eta_2\mu_2}^*(\mathbf{x}^t)\|^2. \tag{78}$$

For $\langle \nabla_2 g_{\eta_2\mu_2}(\mathbf{x}^t, \mathbf{y}^t) - D_{\mathbf{y}}^t, \mathbf{y}_{\eta_2\mu_2}^*(\mathbf{x}^t) - \tilde{\mathbf{y}}^{t+1} \rangle$, we have

$$\langle \nabla_2 g_{\eta_2\mu_2}(\mathbf{x}^t, \mathbf{y}^t) - D_{\mathbf{y}}^t, \mathbf{y}_{\eta_2\mu_2}^*(\mathbf{x}^t) - \tilde{\mathbf{y}}^{t+1} \rangle$$
$$= \langle \nabla_2 g_{\eta_2\mu_2}(\mathbf{x}^t, \mathbf{y}^t) - D_{\mathbf{y}}^t, \mathbf{y}_{\eta_2\mu_2}^*(\mathbf{x}^t) - \mathbf{y}^t \rangle + \langle \nabla_2 g_{\eta_2\mu_2}(\mathbf{x}^t, \mathbf{y}^t) - D_{\mathbf{y}}^t, \mathbf{y}^t - \tilde{\mathbf{y}}^{t+1} \rangle$$
$$\geq -\frac{2}{\mu_g} \|\nabla_2 g_{\eta_2\mu_2}(\mathbf{x}^t, \mathbf{y}^t) - D_{\mathbf{y}}^t\|^2 - \frac{\mu_g}{4} \|\mathbf{y}_{\eta_2\mu_2}^*(\mathbf{x}^t) - \mathbf{y}^t\|^2 - \frac{\mu_g}{4} \|\mathbf{y}^t - \tilde{\mathbf{y}}^{t+1}\|^2 \tag{79}$$

Using the above inequalities, we have

$$\frac{1}{2\tau\beta} \|\mathbf{y}^{t+1} - \mathbf{y}_{\eta_2\mu_2}^*(\mathbf{x}^t)\|^2$$
$$\leq \left(\frac{L_2^g}{2} - \frac{1}{2\beta} + \frac{\mu_g}{4}\right) \|\tilde{\mathbf{y}}^{t+1} - \mathbf{y}^t\|^2 + \left(\frac{1}{2\tau\beta} - \frac{\mu_g}{4}\right) \|\mathbf{y}^t - \mathbf{y}_{\eta_2\mu_2}^*(\mathbf{x}^t)\|^2$$
$$+ \frac{2}{\mu_g} \|\nabla_2 g_{\eta_2\mu_2}(\mathbf{x}^t, \mathbf{y}^t) - D_{\mathbf{y}}^t\|^2$$
$$\leq \left(\frac{3L_2^g}{4} - \frac{1}{2\beta}\right) \|\tilde{\mathbf{y}}^{t+1} - \mathbf{y}^t\|^2 + \left(\frac{1}{2\tau\beta} - \frac{\mu_g}{4}\right) \|\mathbf{y}^t - \mathbf{y}_{\eta_2\mu_2}^*(\mathbf{x}^t)\|^2 + \frac{2}{\mu_g} \|\nabla_2 g_{\eta_2\mu_2}(\mathbf{x}^t, \mathbf{y}^t) - D_{\mathbf{y}}^t\|^2$$
$$\leq \left(\frac{1}{2\tau\beta} - \frac{\mu_g}{4}\right) \|\mathbf{y}^t - \mathbf{y}_{\eta_2\mu_2}^*(\mathbf{x}^t)\|^2 - \left(\frac{3}{8\beta} + \frac{1}{8\beta} - \frac{3L_2^g}{4}\right) \|\tilde{\mathbf{y}}^{t+1} - \mathbf{y}^t\|^2$$
$$+ \frac{2}{\mu_g} \|\nabla_2 g_{\eta_2\mu_2}(\mathbf{x}^t, \mathbf{y}^t) - D_{\mathbf{y}}^t\|^2$$
$$\leq \left(\frac{1}{2\tau\beta} - \frac{\mu_g}{4}\right) \|\mathbf{y}^t - \mathbf{y}_{\eta_2\mu_2}^*(\mathbf{x}^t)\|^2 - \frac{3}{8\beta} \|\tilde{\mathbf{y}}^{t+1} - \mathbf{y}^t\|^2 + \frac{2}{\mu_g} \|\nabla_2 g_{\eta_2\mu_2}(\mathbf{x}^t, \mathbf{y}^t) - D_{\mathbf{y}}^t\|^2 \tag{80}$$

where the first inequality is due to $0 < \tau \leq 1$ and the second inequality is due to $L_2^g \geq \mu_g$; the last inequality is due to $0 < \beta \leq \frac{1}{6L_2^g}$. Then, taking expectation on both sides and rearranging the above inequality, we have

$$\mathbb{E}[\|\mathbf{y}^{t+1} - \mathbf{y}_{\eta_2\mu_2}^*(\mathbf{x}^t)\|^2] \leq \left(1 - \frac{\mu_g\tau\beta}{2}\right) \mathbb{E}[\|\mathbf{y}^t - \mathbf{y}_{\eta_2\mu_2}^*(\mathbf{x}^t)\|^2] - \frac{3\tau}{4} \mathbb{E}[\|\tilde{\mathbf{y}}^{t+1} - \mathbf{y}^t\|^2] + \frac{4\beta\tau\sigma_{\mathbf{y}}^2}{\mu_g BB'} \tag{81}$$

Next, we decompose the $\mathbb{E}[\|\mathbf{y}^{t+1} - y_{\eta_2\mu_2}^*(\mathbf{x}^{t+1})\|^2]$ as follows:

$$\mathbb{E}[\|\mathbf{y}^{t+1} - y_{\eta_2\mu_2}^*(\mathbf{x}^{t+1})\|^2]$$
$$= \mathbb{E}[\|\mathbf{y}^{t+1} - y_{\eta_2\mu_2}^*(\mathbf{x}^t) + y_{\eta_2\mu_2}^*(\mathbf{x}^t) - y_{\eta_2\mu_2}^*(\mathbf{x}^{t+1})\|^2]$$
$$= \mathbb{E}[\|\mathbf{y}^{t+1} - y_{\eta_2\mu_2}^*(\mathbf{x}^t)\|^2] + 2\mathbb{E}[\langle \mathbf{y}^{t+1} - y_{\eta_2\mu_2}^*(\mathbf{x}^t), y_{\eta_2\mu_2}^*(\mathbf{x}^t) - y_{\eta_2\mu_2}^*(\mathbf{x}^{t+1}) \rangle]$$
$$+ \mathbb{E}[\|y_{\eta_2\mu_2}^*(\mathbf{x}^t) - y_{\eta_2\mu_2}^*(\mathbf{x}^{t+1})\|^2]$$

$$\leq \left(1 + \frac{\mu_g \tau \beta}{4}\right) \mathbb{E}[\|\mathbf{y}^{t+1} - y^*_{\eta_2 \mu_2}(\mathbf{x}^t)\|^2] + \left(1 + \frac{4}{\mu_g \tau \beta}\right) L^2_{\mathbf{y}^*_{\eta\mu}} \mathbb{E}[\|\mathbf{x}^t - \mathbf{x}^{t+1}\|^2]$$

$$\leq \left(1 + \frac{\mu_g \tau \beta}{4}\right) \left(\left(1 - \frac{\mu_g \tau \beta}{2}\right) \mathbb{E}[\|\mathbf{y}^t - \mathbf{y}^*_{\eta_2 \mu_2}(\mathbf{x}^t)\|^2] - \frac{3\tau}{4} \mathbb{E}[\|\tilde{\mathbf{y}}^{t+1} - \mathbf{y}^t\|^2] + \frac{4\beta\tau\sigma^2_{\mathbf{y}}}{\mu_g BB'}\right)$$

$$+ \left(1 + \frac{4}{\mu_g \tau \beta}\right) L^2_{\mathbf{y}^*_{\eta\mu}} \mathbb{E}[\|\mathbf{x}^t - \mathbf{x}^{t+1}\|^2]$$

$$= \left(1 + \frac{\mu_g \tau \beta}{4}\right) \left(1 - \frac{\mu_g \tau \beta}{2}\right) \mathbb{E}[\|\mathbf{y}^t - \mathbf{y}^*_{\eta_2 \mu_2}(\mathbf{x}^t)\|^2] - \frac{3\tau}{4} \left(1 + \frac{\mu_g \tau \beta}{4}\right) \mathbb{E}[\|\tilde{\mathbf{y}}^{t+1} - \mathbf{y}^t\|^2]$$

$$+ \left(1 + \frac{\mu_g \tau \beta}{4}\right) \frac{4\tau\beta\sigma^2_{\mathbf{y}}}{\mu_g BB'} + \left(1 + \frac{4}{\mu_g \tau \beta}\right) L^2_{\mathbf{y}^*_{\eta\mu}} \mathbb{E}[\|\mathbf{x}^t - \mathbf{x}^{t+1}\|^2] \tag{82}$$

where the first inequality is due to Young's inequality and Lemma 1. Since $0 < \beta \leq \frac{1}{6L^g_2}$ and $L^g_2 \geq \mu_g$, we have $\beta \leq \frac{1}{6L^g_2} \leq \frac{1}{6\mu_g}$ and $\tau \leq 1 \leq \frac{1}{6\mu_g \beta}$. Then, we have

$$\left(1 + \frac{\mu_g \tau \beta}{4}\right) \left(1 - \frac{\mu_g \tau \beta}{2}\right) = 1 - \frac{\mu_g \tau \beta}{2} + \frac{\mu_g \tau \beta}{4} - \frac{\mu^2 \tau^2 \beta^2}{8} \leq 1 - \frac{\mu_g \tau \beta}{4}, \tag{83}$$

$$- \left(1 + \frac{\mu_g \tau \beta}{4}\right) \frac{3\tau}{4} \leq - \frac{3\tau}{4}, \tag{84}$$

$$\left(1 + \frac{\mu_g \tau \beta}{4}\right) \frac{4\tau\beta\sigma^2_{\mathbf{y}}}{\mu_g BB'} \leq \left(1 + \frac{1}{24}\right) \frac{4\beta\tau\sigma^2_{\mathbf{y}}}{\mu_g BB'} = \frac{25\beta\tau\sigma^2_{\mathbf{y}}}{6\mu_g BB'}, \tag{85}$$

$$\left(1 + \frac{4}{\mu_g \tau \beta}\right) \leq \frac{5}{3}. \tag{86}$$

Therefore, we have

$$\mathbb{E}[\|\mathbf{y}^{t+1} - y^*_{\eta_2 \mu_2}(\mathbf{x}^{t+1})\|^2]$$

$$\leq \left(1 - \frac{\mu_g \tau \beta}{4}\right) \mathbb{E}[\|\mathbf{y}^t - \mathbf{y}^*_{\eta_2 \mu_2}(\mathbf{x}^t)\|^2] - \frac{3\tau}{4} \mathbb{E}[\|\tilde{\mathbf{y}}^{t+1} - \mathbf{y}^t\|^2]$$

$$+ \frac{25\beta\tau\sigma^2_{\mathbf{y}}}{6\mu_g BB'} + \frac{5L^2_{\mathbf{y}^*_{\eta\mu}}}{3} \mathbb{E}[\|\mathbf{x}^t - \mathbf{x}^{t+1}\|^2] \tag{87}$$

$\square$

### D.4 THE DETAILED PROOF OF THEOREM 1

*Proof.* According to Lemma 5, we have

$$\Phi(\mathbf{x}^{t+1}) - \Phi(\mathbf{x}^t)$$

$$\leq - \frac{3\alpha\rho}{8} \|\mathcal{G}^t\|^2 + \frac{16\alpha(L^g_1)^2}{\rho} \|\mathbf{z}^t - \mathbf{z}^*_t\|^2 + \frac{2\alpha}{\rho} A + \frac{4\alpha\sigma^2_{\mathbf{x}}}{\rho BB'}$$

$$+ \frac{2\alpha}{\rho} \left(8(L^f_1)^2 + 8(L^g_2 r_{\mathbf{z}})^2 + 8(L^g_1)^2 \left(\frac{2(L^f_1)^2}{\mu^2_g} + \frac{2(L^f_0 L^g_2)^2}{\mu^4_g}\right)\right) \|\mathbf{y}^t - \mathbf{y}^*_{\eta_2 \mu_2}(\mathbf{x}^t)\|^2$$

$$\leq - \frac{3\alpha\rho}{8} \|\mathcal{G}^t\|^2 + \frac{16\alpha(L^g_1)^2}{\rho} \|\mathbf{z}^t - \mathbf{z}^*_t\|^2 + \frac{2\alpha}{\rho}(A + \frac{2\sigma^2_{\mathbf{x}}}{BB'}) + \frac{2\alpha\hat{L}^2}{\rho} \|\mathbf{y}^t - \mathbf{y}^*_{\eta_2 \mu_2}(\mathbf{x}^t)\|^2 \tag{88}$$

where $\hat{L}^2 = 8(L^f_1)^2 + 8(L^g_2 r_{\mathbf{z}})^2 + 8(L^g_1)^2 \left(\frac{2(L^f_1)^2}{\mu^2_g} + \frac{2(L^f_0 L^g_2)^2}{\mu^4_g}\right)$. According to Lemma 6, we have

$$\mathbb{E}[\|\mathbf{y}^{t+1} - y^*_{\eta_2 \mu_2}(\mathbf{x}^{t+1})\|^2] - \mathbb{E}[\|\mathbf{y}^t - \mathbf{y}^*_{\eta_2 \mu_2}(\mathbf{x}^t)\|^2]$$

$$\leq - \frac{\mu_g \tau \beta}{4} \mathbb{E}[\|\mathbf{y}^t - \mathbf{y}^*_{\eta_2 \mu_2}(\mathbf{x}^t)\|^2] - \frac{3\tau}{4} \mathbb{E}[\|\tilde{\mathbf{y}}^{t+1} - \mathbf{y}^t\|^2] + \frac{25\beta\tau\sigma^2_{\mathbf{y}}}{6\mu_g BB'} + \frac{5L^2_{\mathbf{y}^*_{\eta\mu}}}{3} \mathbb{E}[\|\mathbf{x}^t - \mathbf{x}^{t+1}\|^2]$$

$$\leq -\frac{\mu_g \tau \beta}{4} \mathbb{E}[\|\mathbf{y}^t - \mathbf{y}^*_{\eta_2 \mu_2}(\mathbf{x}^t)\|^2] - \frac{3\tau}{4} \mathbb{E}[\|\tilde{\mathbf{y}}^{t+1} - \mathbf{y}^t\|^2] + \frac{25\beta \tau \sigma^2_{\mathbf{y}}}{6\mu_g BB'} + \frac{5L^2_{\mathbf{y}^*_{\eta \mu}} \alpha^2}{3} \mathbb{E}[\|\mathcal{G}^t\|^2] \quad (89)$$

According to Lemma 4, we have

$$\mathbb{E}[\|\mathbf{z}^{t+1} - \mathbf{z}^*_{t+1}\|^2] - \mathbb{E}[\|\mathbf{z}^t - \mathbf{z}^*_t\|^2]$$

$$\leq -\frac{\mu_g \lambda}{4} \mathbb{E}[\|\mathbf{z}^t - \mathbf{z}^*_t\|^2] - \frac{3}{4} \mathbb{E}[\|\mathbf{z}^{t+1} - \mathbf{z}^t\|^2]$$

$$+ \frac{25\lambda \sigma^2_{\mathbf{z}}}{6\mu_g BB'} + \frac{20}{3}\left(\frac{L^f_1}{\mu_g} + \frac{L^f_0 L^g_2}{\mu_g^2}\right)(\mathbb{E}[\|\mathbf{x}^{t+1} - \mathbf{x}^t\|^2] + \mathbb{E}[\|\mathbf{y}^{t+1} - \mathbf{y}^t\|^2])$$

$$\leq -\frac{\mu_g \lambda}{4} \mathbb{E}[\|\mathbf{z}^t - \mathbf{z}^*_t\|^2] - \frac{3}{4} \mathbb{E}[\|\mathbf{z}^{t+1} - \mathbf{z}^t\|^2]$$

$$+ \frac{25\lambda \sigma^2_{\mathbf{z}}}{6\mu_g BB'} + \frac{20\tilde{L}^2}{3}(\alpha^2 \mathbb{E}[\|\mathcal{G}^t\|^2] + \tau^2 \mathbb{E}[\|\tilde{\mathbf{y}}^{t+1} - \mathbf{y}^t\|^2]) \quad (90)$$

where $\tilde{L}^2 = \frac{L^f_1}{\mu_g} + \frac{L^f_0 L^g_2}{\mu_g^2}$.

Then, we define the following Lyapunov function for any $t \geq 1$,

$$H^t = \mathbb{E}[\Phi(\mathbf{x}^t)] + \mathbb{E}[\|\mathbf{y}^t - \mathbf{y}^*_{\eta_2 \mu_2}(\mathbf{x}^t)\|^2] + \mathbb{E}[\|\mathbf{z}^t - \mathbf{z}^*_t\|^2] \quad (91)$$

By using the above inequalities, we have

$$H^{t+1} - H^t$$

$$\leq -\frac{3\alpha\rho}{8}\mathbb{E}[\|\mathcal{G}^t\|^2] + \frac{16\alpha(L^g_1)^2}{\rho}\mathbb{E}[\|\mathbf{z}^t - \mathbf{z}^*_t\|^2] + \frac{2\alpha}{\rho}\left(A + \frac{2\sigma^2_{\mathbf{x}}}{BB'}\right) + \frac{2\alpha\hat{L}^2}{\rho}\mathbb{E}[\|\mathbf{y}^t - \mathbf{y}^*_{\eta_2 \mu_2}(\mathbf{x}^t)\|^2]$$

$$- \frac{\mu_g \tau \beta}{4}\mathbb{E}[\|\mathbf{y}^t - \mathbf{y}^*_{\eta_2 \mu_2}(\mathbf{x}^t)\|^2] - \frac{3\tau}{4}\mathbb{E}[\|\tilde{\mathbf{y}}^{t+1} - \mathbf{y}^t\|^2] + \frac{25\beta \tau \sigma^2_{\mathbf{y}}}{6\mu_g BB'} + \frac{5L^2_{\mathbf{y}^*_{\eta \mu}} \alpha^2}{3}\mathbb{E}[\|\mathcal{G}^t\|^2]$$

$$- \frac{\mu_g \lambda}{4}\mathbb{E}[\|\mathbf{z}^t - \mathbf{z}^*_t\|^2] - \frac{3}{4}\mathbb{E}[\|\mathbf{z}^{t+1} - \mathbf{z}^t\|^2]$$

$$+ \frac{25\lambda \sigma^2_{\mathbf{z}}}{6\mu_g BB'} + \frac{20\tilde{L}^2}{3}(\alpha^2 \mathbb{E}[\|\mathcal{G}^t\|^2] + \tau^2 \mathbb{E}[\|\tilde{\mathbf{y}}^{t+1} - \mathbf{y}^t\|^2])$$

$$\leq -\left(\frac{3\alpha\rho}{8} - \frac{5L^2_{\mathbf{y}^*_{\eta \mu}} \alpha^2}{3} - \frac{20\tilde{L}^2 \alpha^2}{3}\right)\mathbb{E}[\|\mathcal{G}^t\|^2]$$

$$+ \left(\frac{16\alpha(L^g_1)^2}{\rho} - \frac{\mu_g \lambda}{4}\right)\mathbb{E}[\|\mathbf{z}^t - \mathbf{z}^*_t\|^2]$$

$$+ \left(\frac{2\alpha\hat{L}^2}{\rho} - \frac{\mu_g \tau \beta}{4}\right)\mathbb{E}[\|\mathbf{y}^t - \mathbf{y}^*_{\eta_2 \mu_2}(\mathbf{x}^t)\|^2]$$

$$+ \left(\frac{20\tilde{L}^2 \tau^2}{3} - \frac{3\tau}{4}\right)\mathbb{E}[\|\tilde{\mathbf{y}}^{t+1} - \mathbf{y}^t\|^2]$$

$$+ \frac{2\alpha}{\rho}\left(A + \frac{2\sigma^2_{\mathbf{x}}}{BB'}\right) + \frac{25\beta \tau \sigma^2_{\mathbf{y}}}{6\mu_g BB'} + \frac{25\lambda \sigma^2_{\mathbf{z}}}{6\mu_g BB'} \quad (92)$$

Then, setting $\alpha \leq \min\left\{\frac{3\rho}{4(5L^2_{\mathbf{y}^*_{\eta \mu}} + 20\tilde{L}^2)}, \frac{\rho\mu_g \beta}{8\hat{L}^2}, \frac{\rho\mu_g \lambda}{64(L^g_1)^2}\right\}$ and $0 < \tau \leq \frac{9}{80\tilde{L}^2}$, we have

$$\frac{\alpha\rho}{8}\mathbb{E}[\|\mathcal{G}^t\|^2] \leq H^t - H^{t+1} + \frac{2\alpha}{\rho}\left(A + \frac{2\sigma^2_{\mathbf{x}}}{BB'}\right) + \frac{25\beta \tau \sigma^2_{\mathbf{y}}}{6\mu_g BB'} + \frac{25\lambda \sigma^2_{\mathbf{z}}}{6\mu_g BB'} \quad (93)$$

Taking the average over $t = 1, \cdots, T$ on both sides of the above inequality, we have

$$\frac{1}{T}\sum_{t=1}^{T}\mathbb{E}[\|\mathcal{G}^t\|^2]$$

$$\leq \frac{8(H^1 - H^{T+1})}{\alpha \rho T} + \frac{16}{\rho^2}\left(A + \frac{2\sigma_{\mathbf{x}}^2}{BB'}\right) + \frac{200\beta\tau\sigma_{\mathbf{y}}^2}{6\alpha\rho\mu_g BB'} + \frac{200\lambda\sigma_{\mathbf{z}}^2}{6\alpha\rho\mu_g BB'}$$

$$\leq \frac{8H^1}{\alpha \rho T} + \frac{16}{\rho^2}\left(A + \frac{2\sigma_{\mathbf{x}}^2}{BB'}\right) + \frac{200\beta\tau\sigma_{\mathbf{y}}^2}{6\alpha\rho\mu_g BB'} + \frac{200\lambda\sigma_{\mathbf{z}}^2}{6\alpha\rho\mu_g BB'} \tag{94}$$

$\square$

