# OpenReview forum: "Query Efficient Nonsmooth Stochastic Black-Box Bilevel Optimization with Bregman Distance"
_ICLR.cc/2025/Conference — Submitted to ICLR 2025_

### Official Review · Reviewer_pMgk · 2024-10-23

**Soundness:** 3
**Presentation:** 2
**Contribution:** 3
**Rating:** 5
**Confidence:** 4

**Summary:**

This paper considers nonsmooth black box Bi-level optimization. The lower problem is assumed to be strongly convex and have Lipschitz continuous gradients and hessians, and the upper level function can be nonconvex. Under this setting, the paper improves the query complexity of zero order method for finding an approximately stationary point of BLO. Numerical experiments are conducted to show the competitive performance of the proposed method.

**Strengths:**

1.The paper chooses a good perspective to study BLO methods, where there is still some gaps in the complexity theory.

2.The paper improves the theoretical complexity bound to an almost optimal one.

**Weaknesses:**

1.The strong convexity assumption has been extensively studied for BLO. The authors should lay importance on the difference of the used techniques for establishing complexity bound for black box BLO.

2.The stationarity measure $\|G^t\|$ lack explanation. It is important for the authors to justify that the used measure is equivalent to the compared methods.

**Questions:**

What is the benefit of using the Bregman distance in the proposed algorithm?

---

### Official Review · Reviewer_KDXN · 2024-11-01

**Soundness:** 1
**Presentation:** 2
**Contribution:** 2
**Rating:** 3
**Confidence:** 4

**Summary:**

This paper proposes a query-efficient algorithm, BreZOSBA, designed to solve nonsmooth stochastic black-box bilevel optimization (BO) problems by leveraging Bregman distance and Gaussian smoothing. By adopting a single-loop, zeroth-order (ZO) framework, the authors claim improved query efficiency, theoretically achieving $ O(d_1(d_1 + d_2)^2 \epsilon^{-2})$  query complexity to reach an $ \epsilon$ -stationary point. The paper validates BreZOSBA on two small scale applications—data hyper-cleaning and hyper-representation learning—and compares it to two baselines.

**Strengths:**

- Theoretical Convergence Guarantees: The convergence analysis looks comprehensive, providing non-asymptotic convergence results that are theoretically solid and highlight the advantage of BreZOSBA in terms of query complexity.

- BreZOSBA’s use of a single-loop ZO framework with Bregman distance introduces computational savings compared to double-loop structures.

**Weaknesses:**

- The practical applicability of bilevel optimization with both inner and outer levels as black-boxes was largely not motivated by the authors. While some specific bilevel problems can have partial black-box structure, many practical applications in machine learning (such as hyperparameter optimization, meta-learning, etc) involve inner-level variables that represent the parameters of deep neural networks. In such cases, using zeroth-order methods to solve the inner problem would be extremely slow and may not even converge.

- The choice to employ ZO techniques to solve both the inner and outer levels restricts the scalability of the approach, which may explain why the experimental validation was limited to small-scale toy tasks. Thus, the method’s suitability for real-world BO problems remains questionable, especially in deep learning context.

- The experiments are very limited with only two compared baselines (that are not particularly strong for these experiments). In fact, HOZOG [1] was introduced especially for hyperparameter optimization (where $x$ are the hyperparameters and $y$ is the parameters of possibly a deep neural network), in which it can be a strong baseline. However, the authors used it here in settings for which HOZOG is not the most adequate baseline, which I believe is unfair and undermines the validity of the results.

- The literature review on bilevel optimization is sparse and does not adequately cover recent, relevant works. For instance, PZOBO [2], another hessian-free ZO method, is neither discussed nor compared against, despite its direct relevance.
The paper also lacks a discussion on extensions or limitations of the proposed approach in the broader context of BO, such as bilevel problems without lower-level singleton constraint. This would be essential to clarify where the proposed method fits within the current landscape of BO approaches and where it may fall short or require adaptation.

Other minor issues:

- The inner objective $g$ is smooth and this should be explicitly said after the problem definition, just like the authors did it for the outer objective function $f$.

- In the text, the authors keep mentioning that using ZO gradient for the inner level problem is not the most efficient thing, but their algorithm did exactly use that (equation 14).

References:

[1] Bin Gu, Guodong Liu, Yanfu Zhang, Xiang Geng, and Heng Huang. Optimizing large-scale
hyperparameters via automated learning algorithm. arXiv preprint arXiv:2102.09026, 2021.

[2] On the convergence theory for hessian-free bilevel algorithms. D Sow, K Ji, Y Liang - Advances in Neural Information Processing Systems (NeurIPS), 2022.

**Questions:**

See weaknesses.

---

> ### Comment · Reviewer_KDXN · 2024-12-03
>
> Authors did not bother to acknowledge the efforts made by the reviewers.

---

### Official Review · Reviewer_Y58f · 2024-11-03

**Soundness:** 3
**Presentation:** 3
**Contribution:** 2
**Rating:** 5
**Confidence:** 3

**Summary:**

This paper proposes a gradient-free algorithm, named BreZOSBA, for solving stochastic black-box bilevel optimization problems with nonsmooth upper-level and smooth lower level. To deal with the nonsmoothness and the black-box nature of the problem, the algorithm updates the upper and lower-level variables iteratively using stochastic zeroth-order gradient estimates and a Bregman distance-based proximal term. The theoretical analysis demonstrates the query efficiency of the algorithm, showing that it can achieve a certain accuracy level with a finite number of queries.
The paper also presents experimental results on four datasets (MNIST, FashionMNIST, Cifar10, and SVHN) to evaluate the performance of the proposed method. The results show that BreZOSBA outperforms several baseline methods in terms of accuracy and convergence speed.

**Strengths:**

1.The algorithm BreZOSBA proposed in this paper combines zeroth-order gradient estimations and Bregman distances to solve the challenging black-box bilevel problem. This idea is both novel and inspiring.

2.BreZOSBA can converge to the stationary point within $\mathcal{O}\left(\frac{d_1\left(d_1+d_2\right)^2}{\epsilon^2}\right)$ queries, which is outstanding for the zeroth-order gradient method.

3.The experimental results show that the proposed method outperforms several baseline methods, demonstrating its effectiveness in practical applications.

**Weaknesses:**

1.The subproblem related to updates of $x$ may be difficult for some Bregman distance, especially when the structure of $h(x)$ is unknown or complex.

2.The theoretical convergence analysis of this method relies on several strong assumptions like strong convexity of the lower-level, which may not be met in most practical applications. This can consequently affect the actual performance of the algorithm.

3.Although Gaussian smoothing can approximate the gradient, the additional errors introduced at each iteration may accumulate. Although it can converge to a solution in expectation, it may still be far from the solution after multiple iterations due to a large variance.

**Questions:**

1.I notice that the property of  in the Bregman distance used in the convergence analysis is only the strong-convexity. How does the selection of the Bregman distance function impact the performance of the algorithm? Is there a systematic approach to choosing the optimal function for a given problem? What kind of is used in the numerical experiment?

2.The paper applies Gaussian smoothing approximation techniques to replace the gradient updates. Does the algorithm in this paper have any special structure that supports the feasibility of this technique? Can this technique be adopted by other gradient-based algorithms?

Typo：

1. $\mathcal{G}_t=\frac{1}{\alpha} ( x^t-x^{t+1} )$ if you want to get (64), the coefficient of $\mathcal{B}_{\Psi_t} ( x, x_t )$ in (63) should be $\frac{1}{\alpha}$.

---

### Official Review · Reviewer_oBmF · 2024-11-04

**Soundness:** 2
**Presentation:** 2
**Contribution:** 2
**Rating:** 3
**Confidence:** 4

**Summary:**

This paper proposes a solution for nonsmooth bilevel optimization using Bregman distance, with a focus on zeroth-order gradient approximation via Gaussian smoothing.

**Strengths:**

The paper presents a query-efficient bilevel optimization method tailored for nonsmooth stochastic black-box problems, achieving competitive convergence with lower query complexity and outperforming existing methods in data hyper-cleaning and hyperrepresentation learning tasks.

**Weaknesses:**

The assumption of Lipschitz continuity in Assumption 2 appears inconsistent with strong convexity; further clarification is needed on why this assumption holds in the given setting.

**Questions:**

1. The paper could be divided into two parts: Gaussian smoothing and the Bregman distance method for the nonsmooth outer problem, which currently feel loosely connected. Could the authors further clarify why the nonsmooth problem necessitates the use of zeroth-order gradient descent?
2. The convergence relies heavily on sufficiently large values of  $B$  and $B'$. What happens if  $B$  and $B'$ are of  $\mathcal{O}(1)$? This might provide a fairer comparison with existing methods.
3. In lines 283 and 285, it appears that Gaussian smoothing is applied twice, converting the second-order gradient to first and then to zeroth order. This may introduce significant errors; could the authors elaborate on how they mitigate this error?

I would be open to reconsidering my grade if all of my concerns are addressed.

---

### Meta-Review · Area_Chair_99sG · 2024-12-20

**Metareview:**

This paper studies the nonsmooth stochastic black-box bilevel optimization problems, where the lower-level problem is strongly convex and smooth and the upper-level problem is nonsmooth. Both inner and outer levels are black-boxes problems in the sense that their gradients cannot be efficiently evaluated. a query-efficient method has been proposed to effectively leverage Bregman distance to solve nonsmooth stochastic black-box bilevel optimization problems. Overall, the reviewers find the black-box setting of both lower- and upper-level objectives is less well-motivated, and the combination of Gaussian smoothing and Bregman distance is somewhat artificial. The experiments are alos very limited with only two compared baselines.

Therefore, I suggest authors revise based on the above questions to prepare for the next conference.

**Additional Comments On Reviewer Discussion:**

no response is prpvodef by the authors.

---

### Decision · Program_Chairs · 2025-01-22

Reject